JCB Journal of Cell Biology

**TOOLS**

# An *eGFP-Col4a2* mouse model reveals basement membrane dynamics underlying hair follicle morphogenesis

Duligengaowa Wuergezhen[1,2] , Eleonore Gindroz[1] , Ritsuko Morita[1] , Kei Hashimoto[1,3] , Takaya Abe[4] , Hiroshi Kiyonari[4] , and Hironobu Fujiwara[1,2]

**Precisely controlled remodeling of the basement membrane (BM) is crucial for morphogenesis, but its molecular and tissue-level dynamics, underlying mechanisms, and functional significance in mammals remain largely unknown due to limited visualization tools. We developed mouse lines in which the endogenous collagen IV gene (*Col4a2*) was fused with a fluorescent tag. Through live imaging of developing hair follicles, we reveal a spatial gradient in the turnover rate of COL4A2 that is closely coupled with both the BM expansion rate and the proliferation rate of epithelial progenitors. Epithelial progenitors are displaced with directionally expanding BMs but do not actively migrate on stationary BM. The addition of a matrix metalloproteinase inhibitor delays COL4A2 turnover, restrains BM expansion, and increases perpendicular divisions of epithelial progenitors, altering hair follicle morphology. Our findings highlight the spatially distinct dynamics of BM and their key roles in orchestrating progenitor cell behavior and organ shape during development.**

## Introduction

Multicellular organisms are intricate composites of cells and extracellular matrices (ECMs). ECMs are complex polymer networks of proteins and polysaccharides that provide cells with diverse biochemical and biomechanical cues. One major type of ECM is the basement membrane (BM), which is a thin, dense, sheet-like ECM that surrounds most tissues in metazoans (Yurchenco, 2011). The BM contains a core set of proteins, including laminin, collagen IV, nidogen and perlecan, and many other cell- and tissue-specific BM proteins, BM-modifying proteins, and morphogens (Tsutsui et al., 2021; Yurchenco, 2011). The BM is an evolutionarily ancient ECM that is found across metazoans and is the earliest ECM structure to emerge during development (Fidler et al., 2017; Morrissey and Sherwood, 2015; Smyth et al., 1999).

The BM plays essential roles in animal development and homeostasis by acting as a versatile solid-phase cell-adhesive substrate and signaling platform. By interacting with cells via integrins and other cell surface receptors, this sheet-like ECM regulates various fundamental cell behaviors, including cell adhesion, migration, proliferation, differentiation, apoptosis, polarity, and shape. Moreover, the compositions of BM proteins undergo spatiotemporal specialization during development (Fujiwara et al., 2011, 2018; Manabe et al., 2008). Therefore, the physicochemical properties and organization of the BM dynamically change to orchestrate the fates and behaviors of cells. Deficiencies and mutations in BM genes cause various developmental and homeostatic disorders in multicellular organisms, highlighting the critical role of BMs in the formation and maintenance of multicellular systems (Fidler et al., 2018; Khalilgharibi and Mao, 2021; Lu et al., 2011).

The BM has long been considered a static support structure, such as a floor in a building. However, recent studies suggest that the BM is far more dynamic than previously thought at both the molecular and structural levels (Horne-Badovinac, 2020; Matsubayashi, 2022; Morrissey and Sherwood, 2015; Tsutsui et al., 2021). For example, the molecular turnover of collagen IV is observed during the development of *Drosophila* and *C. elegans* and appears to contribute to proper organ development (Keeley et al., 2020; Matsubayashi et al., 2020; Serna-Morales et al., 2023). Furthermore, the assembly of laminin and collagen IV may affect the mechanical properties of the BM, influencing the architecture of epithelial tissues during tumor progression

[1]Laboratory for Tissue Microenvironment, RIKEN Center for Biosystems Dynamics Research, Kobe, Japan; [2]Graduate School of Medicine, Osaka University, Suita, Japan; [3]Graduate School of Humanities and Sciences, Ochanomizu University, Tokyo, Japan; [4]Laboratory for Animal Resources and Genetic Engineering, RIKEN Center for Biosystems Dynamics Research, Kobe, Japan.

Correspondence to Hironobu Fujiwara: hironobu.fujiwara@riken.jp

R. Morita's current affiliation is Graduate School of Frontier Biosciences, Osaka University, Suita, Japan.



(Fiore et al., 2020). Therefore, perfectly balanced BM remodeling and turnover, which may control spatial and temporal changes in the biochemical and biomechanical properties of the BM, are essential for its dynamic functions. However, the molecular and tissue-level dynamics of the BM, along with their underlying regulatory mechanisms and physiological functions, remain largely unexplored.

A significant challenge in BM biology is examining the cross-scale dynamics of varied, dense, complex supramolecular matrices in vivo, especially in mammals. The primary approach to visualizing BM dynamics involves the genetic tagging of fluorescent molecules to BM proteins. However, certain properties associated with the BM can impede the insertion of a large fluorescent protein without affecting normal BM functions. These properties include (1) large modular structures, (2) specialized intracellular transport and secretory mechanisms, (3) posttranslational processing, (4) the assembly of supramolecular complexes, (5) complex molecular interactions, and (6) unique extracellular physicochemical environments, such as redox status. While a few individuals of *C. elegans* and *Drosophila* have been manipulated to express fluorescently tagged endogenous BM proteins with normal functions (Keeley et al., 2020; Ramos-Lewis et al., 2018), similar approaches in mammals are still being developed. Recently, cells and mice expressing fluorescent fusion proteins for core BM proteins—either endogenous or exogenous—have been generated, significantly contributing to our understanding of BM dynamics (Futaki et al., 2023; Jones et al., 2024; Morgner et al., 2023; Shaw et al., 2020). However, mice homozygous for *mTurq2-Col4a1* and mice homozygous for *Lamb1Dendra2* do not survive, suggesting functional abnormalities in the fusion protein (Jones et al., 2024; Morgner et al., 2023). This highlights the ongoing challenges of generating mice that express endogenous core BM proteins fused with a fluorescent tag while maintaining normal function.

The BM comprises two independent, self-assembling networks of laminin and collagen IV that are interconnected by nidogen and perlecan (Jayadev and Sherwood, 2017). Collagen IV networks are crucial for providing the core structure and tensile strength of BMs (Chlasta et al., 2017; Crest et al., 2017; Haigo and Bilder, 2011; Morrissey and Sherwood, 2015; Pöschl et al., 2004). Collagen IV consists of up to six genetically unique α-chains, which are designated α1(IV)–α6(IV). Each collagen IV polypeptide comprises three distinct domains: a cysteine-rich N-terminal 7S domain of ∼150 amino acids; a central, long, triple-helical collagenous domain of ∼1,300 amino acids; and a globular C-terminal NC1 domain of ∼230 amino acids. Heterotrimeric collagen IV molecules can interact through their 7S domains to form tetramers and vital disulfide bonds between collagen molecules or through their NC1 domains to form dimers and create supramolecular network organizations (Añazco et al., 2016; Duance et al., 1984; Jayadev and Sherwood, 2017; Khoshnoodi et al., 2008; Serna-Morales et al., 2023). Due to their intricate heterotrimeric protein structure and complex intermolecular interactions, various dysfunctional mutations have been reported throughout collagen IV genes, leading to the definition of a broad spectrum of disorders, including embryonic lethality, myopathy, glaucoma, hemorrhagic stroke, nephropathy, and

cochlear dysfunctions (Kashtan et al., 2018; Kuo et al., 2012; Pöschl et al., 2004). Therefore, alterations to the collagen IV amino acid sequence or the introduction of a large fluorescent protein at any position can potentially impede the synthesis, assembly, deposition, and function of collagen IV molecules, posing challenges for fluorescent tagging.

Hair follicle morphogenesis is a complex process involving several key morphogenetic events, such as epithelial stratification, placode development, fibroblast aggregation, bud formation and elongation, and the eventual establishment of a mature, patterned hair follicle architecture and form (Saxena et al., 2019). Genetic studies have demonstrated the critical roles of BM components in these processes (Fujiwara, 2024). However, the dynamic nature of the BM, involving the assembly and disassembly of its components, as well as plasticity, expansion, and movement during organ formation and growth, remains poorly understood, largely due to limitations in current methods for analyzing BM dynamics (Stramer and Sherwood, 2024). Consequently, the mechanisms by which BM dynamics regulate hair follicle development, particularly in terms of their spatiotemporal changes and roles in coordinating morphogenesis events, remain largely unresolved.

In this study, we generated knock-in mouse lines that express fluorescently tagged endogenous collagen IV alpha 2 chains (COL4A2). The mice grew normally and were fertile even when they were homozygous. Using embryonic skin tissues from these mice, we established a 3D live-imaging method to enable the long-term continuous monitoring of COL4A2 within tissues. Furthermore, our study revealed the remarkably dynamic and spatially distinct turnover of collagen IV proteins, which play crucial roles in the expansion of the BM at the tissue level, the orientation of epithelial cell division, and epithelial morphogenesis.

## Results

### Development of a 4D imaging method to visualize basement membrane dynamics in live tissues

To visualize BM dynamics in real-time, we first sought to develop knock-in mice expressing fluorescently tagged endogenous collagen IV. Among the six polypeptide chains of collagen IV, we selected the α2 chain (*Col4a2*), which forms the α1α1α2 heterotrimer and is present in the BM of all tissues (Khoshnoodi et al., 2008). By referring to the insertion site of the GFP protein in the viking$^{G454}$ *Drosophila* collagen IV GFP-trap line (Morin et al., 2001), a cDNA encoding the eGFP protein with short linker sequences was inserted into the 7S domain of the collagen IV α2 chain (between $A_{28}$ and $Q_{29}$) via the CRISPR/Cas9-assisted knock-in method in mouse zygotes (Abe et al., 2020) (Fig. 1 A and Fig. S1 A). We confirmed successful transgene insertion into the *Col4a2* gene in *eGFP-Col4a2* knock-in pups with PCR screening and genomic sequencing (Fig. S1, B and C). Importantly, the *eGFP-Col4a2* allele was transmitted at the expected Mendelian ratio of 1:2:1 when knock-in heterozygous mice were bred (Fig. 1 B and Table 1). Furthermore, adult homogeneous knock-in mice were visually indistinguishable from their wild-type littermates (Fig. 1 C). These results indicated that

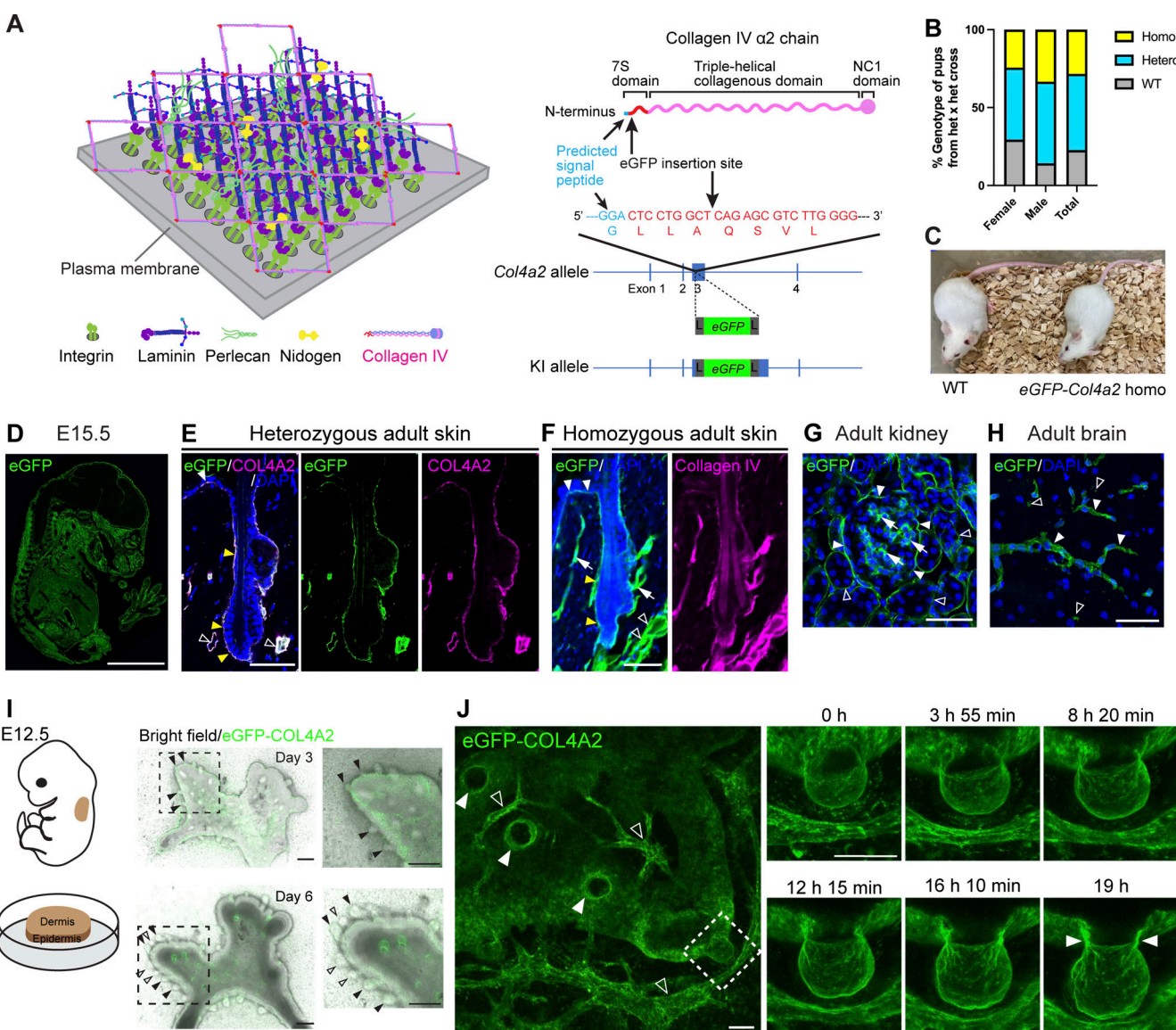

Figure 1. **Development of a 4D imaging method for visualizing basement membrane dynamics. (A)** Schematic model of BM molecular structure and fluorescent protein insertion site in the collagen IV α2 chain. **(B)** Genotypes of pups from *eGFP-Col4a2* het × *eGFP-Col4a2* het crosses. Sample sizes in Table 1. **(C)** Appearance of 8-wk-old wild-type and *eGFP-Col4a2* homo mice. **(D)** Immunofluorescence image of E15.5 embryos from *eGFP-Col4a2* mice stained for eGFP. Scale bar: 3 mm. **(E–H)** Immunofluorescence images of adult tissues from *eGFP-Col4a2* mice. **(E)** In P56 adult telogen dorsal skin from het mice, eGFP (green) colocalized with COL4A2 (magenta) in BMs of epidermis (white arrowheads), follicle epithelium (yellow arrowheads), and vessel-like tissues (open arrowheads). **(F)** In homo mice, eGFP (green) colocalizes similarly with collagen IV (magenta) and was also found in arrector pili muscles (arrows). The same field of view is presented in Fig. S1 I. **(G)** In adult kidneys, eGFP (green) was detected in Bouman's capsules (closed arrowheads), the mesangial matrix (arrows), and collecting ducts (open arrowheads). **(H)** In adult brain, eGFP (green) localized to capillaries (closed arrowheads) and fractone-like structures (open arrowheads). DAPI (blue) = nuclear counterstaining. Scale bars: 50 µm. **(I)** Fluorescence images of cultured dorsal skin explants from E12.5 embryos of *eGFP-Col4a2* mice. Magnified areas show primary (closed arrowheads) and secondary (open arrowheads) hair follicles. Scale bars: 200 µm. **(J)** Snapshot images of the 3D maximum projection of cultured E12.5 skin explants from *eGFP-Col4a2* mice. Closed arrowheads indicate ring-like accumulations of the eGFP-COL4A2 signal at the neck regions of hair follicles. Hair follicles, indicated by closed arrowheads in the left panel, grow vertically from the plane of observation. The dashed rectangle indicates a follicle growing horizontally. The snapshot images in the right panels were obtained from a separate scan of the follicle outlined by the dashed rectangle in the 3D time-lapse maximum projection shown in the main image. The open arrowheads indicate vessel-like structures. Scale bars: 50 µm.

heterozygous and homozygous *eGFP-Col4a2* knock-in mice developed normally and were fertile.

To examine whether the fluorescent fusion protein was appropriately incorporated into the BMs, we immunohistochemically detected eGFP in the embryonic and adult tissues of *eGFP-Col4a2* knock-in mice. eGFP was detected throughout the embryos at E13.5, E15.5, and E17.5, as evidenced by a BM-like linear staining pattern at the tissue borders of many organs, including the skin, kidneys, and brain (Fig. 1 D and Fig. S1, D–G). In the adult skin of *eGFP-Col4a2* heterozygous mice, eGFP colocalized with COL4A2 and perlecan in the BMs of epidermal, hair follicle epithelial, arrector pili muscle, and vessel-like tissues

**Table 1.** Genotypes of pups from *eGFP-Col4a2* het × *eGFP-Col4a2* het crosses

| Genotype | WT | Heterozygous | Homozygous | Total |
|---|---|---|---|---|
| Number of pups born (female:male) | 32 (23:9) | 69 (36:33) | 40 (19:21) | 141 (78:63) |
| Ratio of genotype occurrence | 22.7% | 48.9% | 28.4% | |

(Fig. 1 E and Fig. S1 H), suggesting that the eGFP tags were fused with COL4A2 proteins and that the fusion proteins were localized correctly in the BMs. In homozygous mice, antibodies against eGFP, pan-collagen IV, and perlecan showed BM-like staining patterns and colocalized with each other (Fig. 1 F and Fig. S1 I), suggesting that normal BMs were formed even when both endogenous *Col4α2* alleles were replaced by the knock-in alleles. In adult kidneys, eGFP was detected in the glomeruli along the Bowman's capsule and collecting ducts and in an amorphous pattern in the mesangial matrix (Fig. 1 G and Fig. S1 J), as described in a previous report (Miner and Sanes, 1996). In the adult brain, eGFP overlapped with perlecan and localized in blood vessels, where the blood–brain barrier forms, and in fractone-like structures (Fig. 1 H and Fig. S1 K). We concluded that the eGFP-COL4A2 fusion protein was appropriately incorporated into the BMs and functioned normally.

We then examined whether the eGFP-COL4A2 fusion protein had a fluorescence intensity that would be sufficient to visualize BM dynamics in long-term live imaging of 3D tissues. We employed an ex vivo explant culture system of embryonic dorsal skin that recapitulated the development of hair follicles in vivo (Fig. 1 I) (Morita et al., 2021). Confocal 3D live imaging captured spatiotemporal changes in eGFP-COL4A2. For example, changes in the shape of BMs were captured in developing hair follicles and vessels without significant photobleaching (Fig. 1, I and J; and Video 1). Moreover, a ring-like accumulation of eGFP-COL4A2 was observed in the neck region of the developing hair follicles (Fig. 1 J, closed arrowheads). These results demonstrate that the production of knock-in mice expressing endogenously tagged fluorescent COL4A2 is a viable approach for studying BM dynamics in situ.

**Spatially distinct basement membrane expansion rates synchronize with directional cell movement**

To investigate how BMs spatiotemporally expand in concert with elongating and shape-changing developing organs, we first bleached eGFP fluorescence in several rectangular regions of the BM to compartmentalize BM zones and measured the changes in the length of these BM compartments (Fig. 2, A–C). To maintain the photobleached edges, we conducted repetitive photobleaching cycles every 2–3 h (Materials and methods and Fig. S2). After 7 h of culture, the BM showed an ~30% increase in length at the tip of the growing hair follicle, and the rate of elongation gradually decreased toward the upper part of the hair follicle, with no or reduced elongation in the upper stalk region (Fig. 2, B and C). Therefore, the BM of developing hair follicles does not expand uniformly, similar to a passive balloon inflating, but rather undergoes spatially distinct expansion, generating a spatial gradient of expansion rates.

Considering the rapid expansion of the BM near the tip, we explored the structural differences of the BM in different regions. We used the eGFP-COL4A2 signal as a proxy for the BM structure and first measured its thickness in different tissue regions. The BM at the junction/upper stalk region was 55% and 53% greater than that at the tip and lower stalk regions, respectively (Fig. 2, D and E). While the extensive mesh-like microperforations have been detected in the embryonic mouse salivary glands, lungs, kidneys, and pre-gastrulation embryos (Harunaga et al., 2014; Kyprianou et al., 2020), we did not detect these typical microperforations (Fig. S3 A). Instead, there were instances where pores were present in the BM of the lower part of some developing hair follicles (Fig. S3 B). We assume that these might be the temporal pores for the transmigration of cells such as melanocytes and immune cells through the BM (Adameyko et al., 2009; Bahr et al., 2022; Kabashima et al., 2019; Vandamme and Berx, 2019). These pores were not observed after treatment with a matrix metalloproteinase inhibitor (batimastat) (Fig. S3 C). These observations suggest organ-specific variations in BM remodeling.

BMs can regulate tissue shape by guiding changes in cellular behavior or creating patterned physical constrictions (Morrissey and Sherwood, 2015). We next examined the relationships between the BM expansion rate and the proliferation of basal epithelial progenitors. EdU incorporation assays revealed that basal epithelial progenitors were highly proliferative in the tip region, where high BM expansion was observed, and this percentage gradually decreased toward the upper part of the hair follicle (Fig. 2, F and G), suggesting that the rate of cell proliferation is associated with the degree of BM expansion.

We further examined cellular movements on this expanding BM and considered how much of the observed cell movement could be attributed to the relative displacement of cells against the underlying BM and how much resulted from coordinated movement with the BM. To track the movement of epithelial basal progenitors and their relative position to the underlying BM, we bleached eGFP-COL4A2 fluorescence in several regions of the BM and measured changes in both the cell position and length of the BM (Fig. 2 H). Over 9.5 h, cells in the upper stalk region were displaced ~4% toward the tip of the hair follicle from the reference point on the BM (bleached upper edge), while cells in the lower stalk were displaced ~32% (Fig. 2 I and Fig. S4). The underlying BM shrank ~3% in the upper stalk and elongated ~22% in the lower stalk toward the tip of the hair follicle. To evaluate the contribution of BM expansion to the overall cell displacement, we subtracted the contribution of BM expansion from the cell displacement (Fig. 2 J). In the lower stalk region, the contribution of BM expansion to total cell displacement was ~69% (Fig. 2 J). These findings indicate coordinated directional movement of both cells and the BM toward the tip of the hair follicle. Our measurement results further support this finding by showing that the bleached edge of the BM moved

Figure 2. **Spatially distinct basement membrane expansion rates synchronize with directional cell movement. (A)** Schematic of region definitions in the developing hair follicle. Tip: Interface region between the prematrix and dermal condensate, defined by two bending points at the follicle tip. Lower stalk: The lower half of the follicle extends to the tip. Upper stalk: The upper half extends to the junction region. Junction: Bending neck region where follicular epidermis meets interfollicular epidermis. BM dynamics were measured in these regions. **(B)** Confocal images of hair follicles in dorsal skin explants from *eGFP-Col4a2* mice show that BM length changes over 7 h. ROIs (dashed rectangles) indicate photobleached areas. Re-photobleaching of the targeted BM was performed every 2–3 h to maintain clear photobleached edges. See Fig. S2 for details. BM length changes were measured at the tip (blue lines), lower stalk (orange lines), and upper stalk (gray lines). Scale bar: 20 μm. **(C)** Percent BM length changes over a 7-h (*n* = 9 follicles from seven explants, each from an independent experiment). Values: means ± SD. One-way ANOVA with Tukey's post-hoc tests was used. **(D)** Measurement of BM thickness in hair follicles of dorsal embryonic skin explants from *eGFP-Col4a2* mice. BMs at the tip (blue lines), lower stalk (orange lines), and upper stalk (gray lines) were measured. Scale bars: 20 μm (left two images), 2 μm (right three images). **(E)** Quantification of BM thickness (*n* = 13 follicles from three explants, each from an independent experiment). Values: means ± SD. One-way ANOVA with Tukey's post-hoc tests was used. **(F)** EdU incorporation assays. Dividing cells were labelled with EdU (magenta) with Hoechst counterstaining (blue). EdU-positive nuclei in the basal layer of the tip (blue line), lower stalk (orange line), and upper stalk (gray line) were measured across z-planes (–10, 0, +10, and +15 μm). Scale bar: 20 μm. **(G)** Percentage of EdU-positive nuclei in the tip, lower stalk, and upper stalk

regions (n = 18 follicles from seven explants, each from an independent experiment). Values: means ± SD. One-way ANOVA with Tukey's post-hoc tests was used. **(H)** Snapshot images of 3D time-lapse videos of developing hair follicles in *eGFP-Col4a2; mem-tdTomato* explants. The relative locations of the BM (lower bleached edge) and epithelial basal progenitors (white arrowheads) from reference points on the BM (upper bleached edge, yellow arrowheads) in lower (orange lines) and upper stalk regions (gray lines) were measured at 0 and 9 h 30 min. Re-photobleaching of the targeted BM was performed. See Fig. S4 for detailed tracking procedures. Scale bar: 20 μm. **(I)** Normalized length changes in cell displacement and BM expansion (n = 7 cells and BM regions from five follicles, each from a separate explant in an independent experiment). Values: means ± SD. Two-tailed unpaired *t* tests were used. **(J)** Normalized length changes (relative to cell displacement) of BM expansion and the value subtracted BM expansion from the cell displacement (data from Fig. 2 I). Values: means ± SD. Two-tailed unpaired *t* tests were used. **(K)** A schematic of the composite cell and BM movement. All statistical tests in this figure assumed normal data distribution, but this was not formally tested.

---

alongside the cells (Fig. 2 H and Fig. S4). Taken together, these results suggest that basal epithelial cells move in unison with the directionally expanding underlying BM but do not actively migrate on the stationary BM (Fig. 2 K).

**Increased basement membrane expansion is coupled with increased turnover of collagen IV protein**

The molecular mechanism that facilitates the expansion of pre-existing, edgeless BMs is largely unknown. This may be driven by the incorporation of new collagen IV protein through homeostatic remodeling of the existing collagen IV network. However, how much incorporation and turnover of BM molecules occur spatiotemporally during morphogenesis has not been quantified at the cellular level. Therefore, we measured the dynamics of eGFP-COL4A2 fluorescence recovery after photobleaching (FRAP) as a proxy for the incorporation of the collagen IV protein into the BM (Fig. 3 A and Video 2). Surprisingly, unlike the slow turnover rates of collagens reported in adult animals (from days to months) (Price and Spiro, 1977; Verzijl et al., 2000; Walker, 1973), the fluorescence at the tip region of the hair follicle recovered by 50% just at ~3 h 25 min (205 min), as shown by our fitting curve data, after bleaching (Fig. 3, A and B), indicating a remarkably rapid COL4A2 incorporation rate. Fluorescence recovery was evenly observed across the large bleached area but did not occur from its edges, suggesting that the recovery of fluorescence occurred through the incorporation of eGFP-COL4A2 from outside the BM rather than via diffusion from adjacent unbleached BM areas. In contrast, eGFP-COL4A2 in the lower stalk and the junction regions recovered by about ~20.3% and ~5.7%, respectively (Fig. 3 B), indicating a relatively slow COL4A2 incorporation rate. These observations are supported by statistical tests with our actual measurement data, which show that at 3 h 30 min after bleaching, the fluorescence at the tip region had recovered by 54%, whereas a slow recovery, 23% and 7%, was observed at the lower stalk and junction regions, respectively (Fig. 3 C).

Furthermore, to monitor the dynamics of the COL4A2 protein present at specific locations and time points, we created a knock-in mouse line named *mKikGR-Col4a2*, which expresses endogenous COL4A2 fused with the photoconvertible fluorescent protein mKikGR. This mouse model allows us to measure the dynamics of both pre-existing mKikGR-COL4A2 protein (photoconverted to red (magenta) and newly incorporated mKikGR-COL4A2 proteins (green) in a specific region over time. In the lower stalk region of the BM, the magenta signal gradually decreased and almost disappeared during 8–10 h of culture, whereas the green fluorescence gradually increased and

replaced the red (magenta) signal (Fig. 3 D; and Video 3; and Fig. S5). This red (magenta)-to-green color change indicated turnover of the COL4A2 protein, where pre-existing mKikGR-COL4A2 proteins were replaced by newly synthesized or recruited proteins. In contrast, in the upper stalk and the follicular–interfollicular junction regions, photoconverted red (magenta) mKikGR-COL4A2 remained. These results suggest that turnover of the COL4A2 protein occurs in the BM, rather than mere incorporation or diffusion from surrounding areas. This mouse model's low fluorescence intensity, low signal-to-noise ratio, and high susceptibility to photobleaching limit its use for quantifying molecular dynamics in our system.

**Matrix metalloproteinases are required for COL4A2 incorporation, basement membrane expansion, and hair follicle morphogenesis**

The proteolytic cleavage of ECM components plays essential roles in ECM remodeling (Lu et al., 2011). At the molecular level, proteolysis of the BM components may nick the assembled BM proteins, which may stimulate their degradation and metabolism, rendering the BM more pliable and creating new insertion sites for free BM proteins. This process can facilitate the metabolism or turnover and remodeling of the BM. The central enzymes for ECM remodeling are matrix metalloproteinases (MMPs) (Lu et al., 2011). Recent advancements in ECM imaging have begun to reveal that protease activity in MMPs is required for BM molecular and tissue-level dynamics, as well as effective branching morphogenesis and early embryonic development (Harunaga et al., 2014; Keeley et al., 2020; Matsubayashi et al., 2020). Therefore, we hypothesized that MMP activity is required to generate the observed BM dynamics in hair follicle morphogenesis at the molecular and tissue levels. We tested this by treating skin explants with a broad-spectrum peptidomimetic MMP inhibitor, batimastat (BB-94), which binds to the active sites of MMPs and inhibits the enzymatic activities of several MMPs, including MMP-2 and MMP-9, which cleave collagen IV (Monaco et al., 2006). After 16 h of treatment with the MMP inhibitor (Fig. 4 A), the fluorescence recovery of eGFP-COL4A2 in the FRAP assays was greatly delayed in all regions of hair follicle BMs (Fig. 4, B–D and Video 4). The fluorescence in the tip region of the control hair follicles recovered by 50% ~3 h 30 min after bleaching, whereas the inhibited samples recovered by only ~5% (Fig. 4, C–D). Similarly, the recovery rates of the lower stalk and junction regions drastically decreased (Fig. 4 D). These reduced recovery rates under the MMP inhibitor treatment were not due to the inhibition of the expression of *Col4a2* mRNA (Fig. S6, A and B). These quantitative data indicate that the

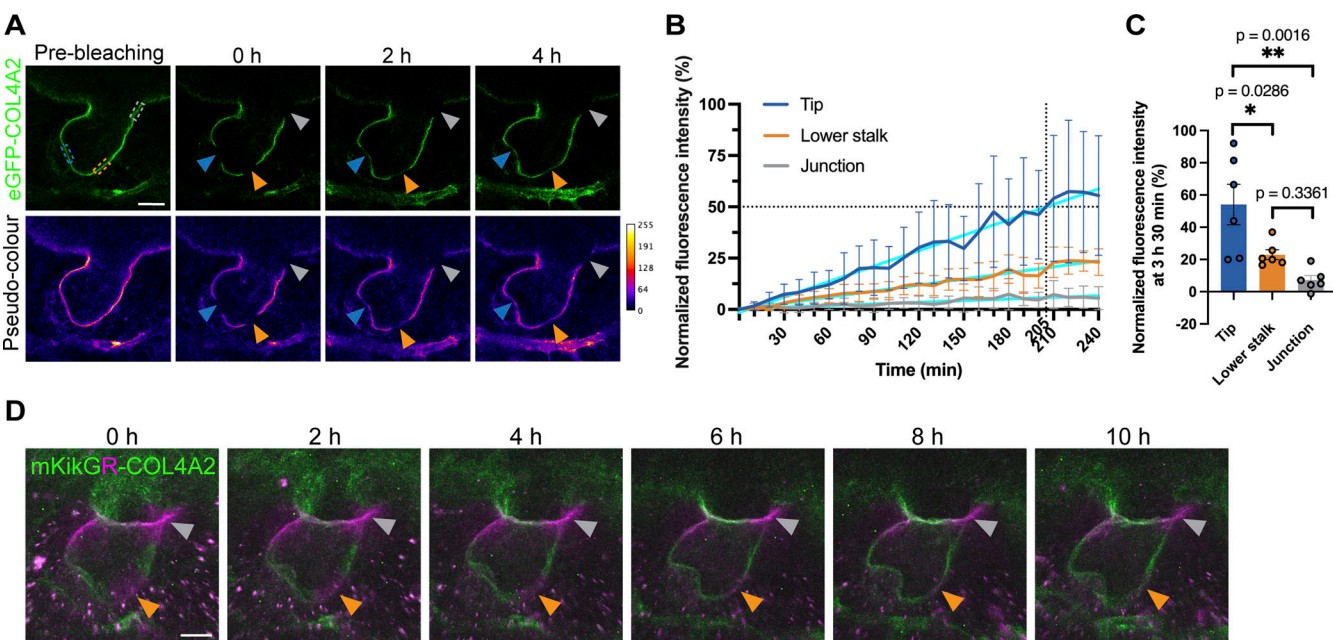

Figure 3. **Spatially distinct collagen IV turnover. (A)** Time-lapse images of eGFP-COL4A2 FRAP experiments. Confocal single-color images (upper) and pseudocolor images (lower) show the BM of a developing hair follicle in explants that were photobleached in the tip (blue arrowheads), lower stalk (orange arrowheads), and follicular–interfollicular junction (gray arrowheads) regions before and after photobleaching at selected recovery times. Scale bar: 20 μm. **(B)** Line graph of the normalized fluorescence recovery of eGFP-COL4A2 over 4 h (n = 6 hair follicles, each from a separate explant in an independent experiment). The cyan lines are simple linear regressions for the data derived from each region. Values: means ± SD. **(C)** Comparison of the recovery rates of eGFP-COL4A2 signals between BMs in different regions. The normalized mean intensity values of eGFP-COL4A2 fluorescence in the bleached regions from the FRAP experiments at 3 h and 30 min, as shown in Fig. 3 B, were used. Values: means ± SD. One-way ANOVA with Tukey's post-hoc tests was used. Data distribution was assumed to be normal, but this was not formally tested. **(D)** Maximum projection images of mKikGR-COL4A2 in developing hair follicles in embryonic skin explants from *mKikGR-Col4a2* mice at the indicated times after photoconversion. Scale bar: 20 μm.

enzymatic activities of MMPs are crucial for COL4A2 incorporation into BMs.

We then investigated changes in the BM expansion rate in MMP inhibitor–treated hair follicles. We measured BM expansion during two distinct time periods: the first 7 h immediately following the addition of MMP inhibitor and another 7 h starting 16 h after the addition of MMP inhibitor. Quantitative analysis revealed that, although not statistically significant, with the MMP inhibitor, the BM showed different expansion patterns during the first 7-h period: the BM in the tip region tended to exhibit a decreased expansion, while the BM in the lower and upper stalk regions tended to shrink further (Fig. 4 E). However, during the second 7-h period (16–23 h after the addition of the MMP inhibitor), the tip region only showed an ∼6% increase, not the ∼30% increase observed in the control (Fig. 4 F). There was almost no elongation or shrinkage in the other regions. These results indicate that the enzymatic activities of MMPs are required for COL4A2 incorporation and BM expansion. While an early stage of MMP inhibition or partial MMP inhibition allows some BM expansion, the spatial pattern of BM expansion is altered. Moreover, our findings suggest a strong link between collagen IV turnover and BM expansion, as indicated by the correlation observed under control conditions. The fact that both COL4A2 turnover and BM expansion are inhibited under MMP inhibition provides additional evidence supporting this relationship, highlighting the critical role of turnover in driving BM expansion and establishing spatial gradients in BM dynamics.

Alterations in BM dynamics and structure could significantly affect organ morphogenesis and shape (Fiore et al., 2020; Ramos-Lewis and Page-McCaw, 2019). Therefore, we examined organ shape changes in developing hair follicles from the hair germ to hair peg stages with and without an MMP inhibitor. During the early stage of MMP inhibitor treatment (0–16 h), hair follicles exhibited an abnormal, disproportionate shape (Fig. 4 G). Inhibitor-treated hair follicles showed halted hair follicle elongation (denoted as $L_{HF}$; Fig. 4, H–J) and increased hair follicle width (denoted as $W_{HF}$; Fig. 4, H, K, and L). We describe these distinct tissue architectures as a shape factor, S, defined as the ratio of $L_{HF}$ to $W_{HF}$ (Fig. 4 H). We detected high and increasing S values in control hair follicles, indicating the formation of an elongating cylindrical structure during normal development (Fig. 4, M–O). In contrast, the S value of the inhibitor-treated hair follicles did not increase as development progressed, and the S value even tended to decrease (Fig. 4, M–O), indicating the formation of a widened hair follicle structure compared with that of the control. This widening effect during the early stage of MMP inhibition, when BM expansion is still possible, suggests that partial BM expansion with an altered spatial pattern leads to an abnormal follicle shape. However, during the later stage of MMP inhibitor treatment, where the incorporation of collagen IV protein was greatly retarded and BM expansion was almost completely inhibited (Fig. 4, B–D and F), hair follicles did not undergo further shape changes under MMP inhibition compared with those in the

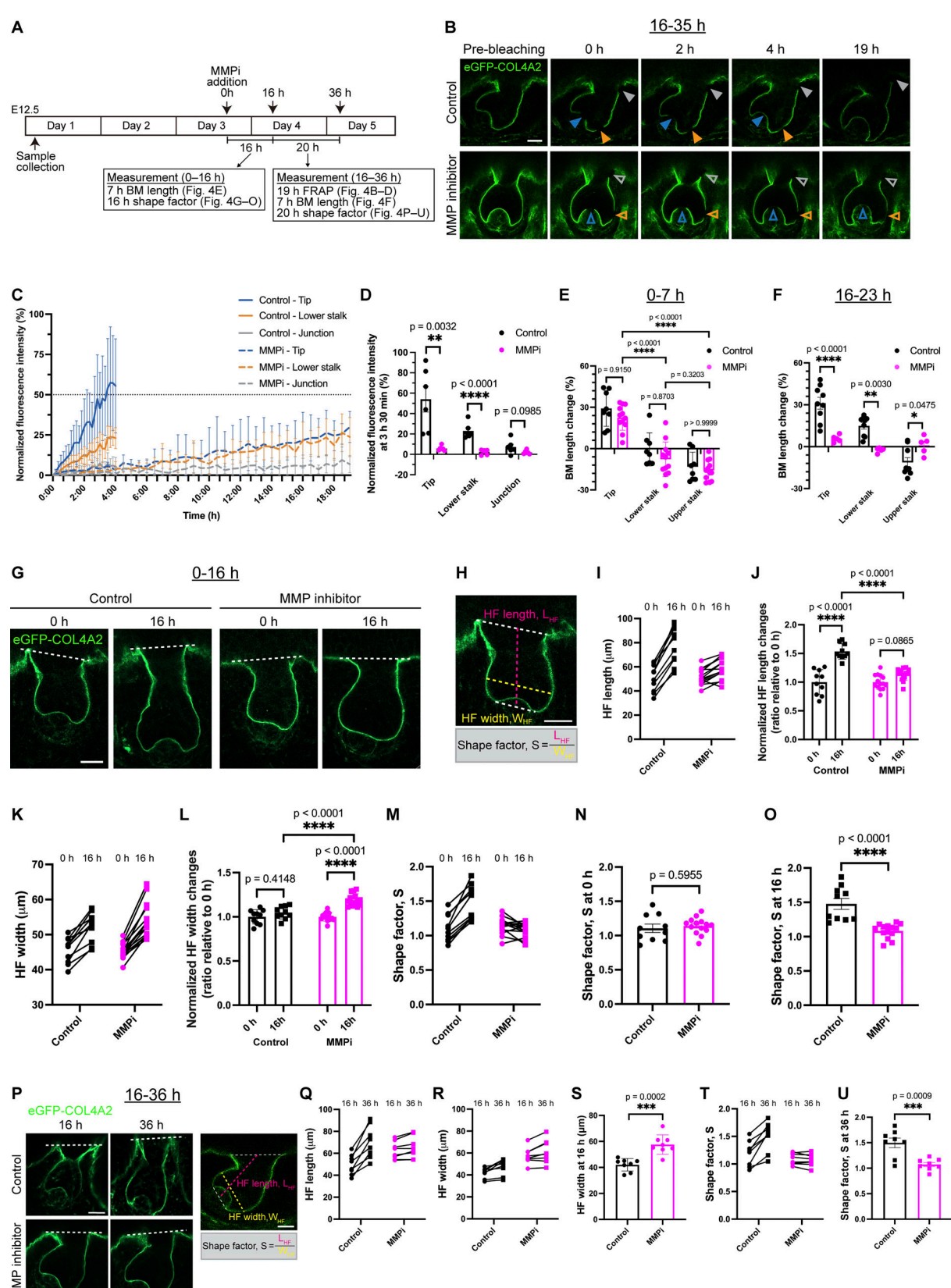

Figure 4. **Matrix metalloproteinases are required for COL4A2 incorporation, basement membrane expansion and hair follicle morphogenesis.** **(A)** Schematic of the experimental procedure for MMP inhibition. **(B)** Snapshot images of eGFP-COL4A2 FRAP experiments in follicles from *eGFP-Col4a2* explants, with or without MMP inhibitor. eGFP-COL4A2 (green) recovery after photobleaching is shown at the tip (blue arrowheads), lower stalk (orange arrowheads), and junction (gray arrowheads) regions in control and MMP-inhibited follicles (open arrowheads). Scale bar: 20 μm. **(C)** Normalized mean

intensity values of eGFP-COL4A2 in bleached regions ($n$ = 6 follicles, each from a separate explant in an independent experiment for the control, and $n$ = 6 follicles from four explants, each from an independent experiment for the inhibitor-treated condition). Values normalized to 0 h. MMPi = MMP inhibitor treatment. Values: means ± SD. The control data of this experiment are shown in Fig. 3 B. **(D)** Quantification of the mean intensity at 3 h 30 min in C. Values: means ± SD. Two-tailed unpaired $t$ tests were used. **(E)** BM length changes during the early stage of MMP inhibition (0–7 h) ($n$ = 8 follicles from four explants, each from an independent experiment for the control, and $n$ = 12 follicles from four explants, each from an independent experiment for the inhibitor-treated). Re-photobleaching of the targeted BM was performed every 2–3 h. Values: means ± SD. Two-way ANOVA with Tukey's post-hoc tests was used. **(F)** BM length changes during the later stage of MMP inhibition (16–23 h) ($n$ = 9 follicles from six explants, each from an independent experiment for the control, and $n$ = 5 follicles from three explants, each from an independent experiment for the inhibitor-treated). Re-photobleaching of the targeted BM was performed. Values: means ± SD. Two-tailed unpaired $t$ tests were used. The control data of this experiment are shown in Fig. 2 C. **(G–O)** Morphological changes in follicles during the early stage of MMP inhibition (0–16 h). **(G)** Confocal images. **(H)** Shape factor: S = follicle length ($L_{HF}$)/bulb width ($W_{HF}$). Changes in the length of follicles (I), the normalized length (J), width (K), normalized width (L), shape factor changes (M), shape factor at 0 h (N), and at 16 h (O) ($n$ = 10 follicles from five explants, each from an independent experiment for the control, and $n$ = 14 follicles from four explants, each from an independent experiment for the inhibitor-treated). The values were normalized to 0 h (J and L). Values: means ± SD. Two-tailed unpaired $t$ tests were used. Two-way ANOVA with Tukey's post-hoc tests was used for J and L. Scale bars: 20 µm. **(P–U)** Morphological changes during the later stage of MMP inhibition (16–36 h). **(P)** Confocal images. Changes in follicle length (Q), width (R), width at 16 h (S), shape factor changes (T), and shape factor at 36 h (U). ($n$ = 8 follicles from seven explants, each from an independent experiment for the control, and $n$ = 8 follicles from five explants, each from an independent experiment for the inhibitor-treated). Values: means ± SD. Two-tailed unpaired $t$ tests were used. Scale bars: 20 µm. All statistical tests in this figure assumed normal data distribution, but this was not formally tested.

control during this period (Fig. 4, P–U). These observations suggest that MMP inhibitor treatment initially allows BM expansion with an altered pattern, leading to the widening of the hair follicle, but in the later stage, both BM expansion and hair follicle shape changes are inhibited. We conclude that the enzymatic activities of MMPs are required to incorporate and turnover COL4A2 proteins, as well as to expand the BM. The inhibition of these activities disrupts hair follicle morphogenesis.

### Inhibiting matrix metalloproteinases alters the orientation of the daughter cell allocation of epithelial progenitors

The spatiotemporal coordination of cell proliferation and the spatial arrangement of cells is one of the major determinants of organ shape (Baena-López et al., 2005). To investigate the potential causes of the wider shape of developing hair follicles after the administration of the MMP inhibitor, we examined the cell division pattern of epithelial basal progenitors in live images (Fig. 5 A). During the 9.5-h imaging period, basal cell division slightly, but not significantly, decreased in inhibitor-treated hair follicles (Fig. 5 B). In the EdU incorporation assays, the number of EdU-positive basal cells decreased at the tip and lower stalk regions in the inhibitor-treated hair follicles, whereas no notable decrease was observed in the upper stalk region (Fig. 5, C and D), where the effects of the MMP inhibitor on BM dynamics were small. These data indicate that although MMP inhibition reduced the proliferation of basal cells, a significant number of basal cells continued to proliferate within our experimental period.

We used our live imaging data to further examine the angle of postmitotic daughter cell allocation compared with that of the BM. In the control hair follicles, 50.0% of the divided cells formed cell allocation angles lower than 30% ($\theta < 30°$; horizontal) and 39.6% formed cell allocation angles of >60% ($\theta > 60°$; perpendicular) (Fig. 5, A and E–G; and Video 5). Strikingly, with the MMP inhibitor, only 13.9% of the divided daughter cells were allocated to horizontal angles and 75.0% showed perpendicular allocation (suprabasal direction) (Fig. 5 F). This shift was observed in all the tissue regions (Fig. 5 G). Therefore, the orientation of the daughter cell allocation of basal epithelial

progenitors is tightly linked to MMP-dependent COL4A2 protein dynamics and BM expansion. These results suggest that the increased cell supply toward the center of epithelial tissue in the tip and lower stalk regions contributed to generating the wider hair follicle tissue shape observed under the inhibition of MMP activity.

## Discussion

In this study, we successfully generated knock-in mice that express fluorescence-tagged endogenous COL4A2. Through live imaging and quantitative analysis of COL4A2 dynamics in hair follicle development, we revealed notable spatial gradients in the COL4A2 turnover rate and the BM expansion rate (Fig. 6). Importantly, these two processes exhibit a coupled relationship. Furthermore, epithelial progenitors move together with the expanding BM, and the inhibition of MMPs substantially suppresses both COL4A2 turnover and BM expansion. This inhibition almost completely shifts the division angle of epithelial progenitors toward the perpendicular direction. Moreover, epithelial tube elongation halts and a wider tube shape emerges. Our findings elucidate the interplay between COL4A2 turnover, BM expansion, and epithelial progenitor behavior, illuminating the complex orchestration of tissue morphogenesis via the molecular and functional dynamics of the BM.

### The importance and challenges of fluorescence tagging of endogenous basement membrane proteins

A major challenge in ECM biology is quantitatively examining the dynamics of supramolecular assembled complex matrices in situ at both the molecular and tissue levels. While fluorescence-labeled antibodies to ECM proteins have been valuable for visualizing ECM dynamics (Bardsley et al., 2017; Harunaga et al., 2014; Mao and Schwarzbauer, 2005), they have limitations, including the possible masking of active epitopes, steric hindrance, variable efficiency of ECM protein labeling across time and space, and selective binding to specific protein forms (Engvall et al., 1986; Nagai et al., 1991). Injecting fluorescently labeled ECM proteins present similar challenges as the proximity and activity of the added protein may affect the

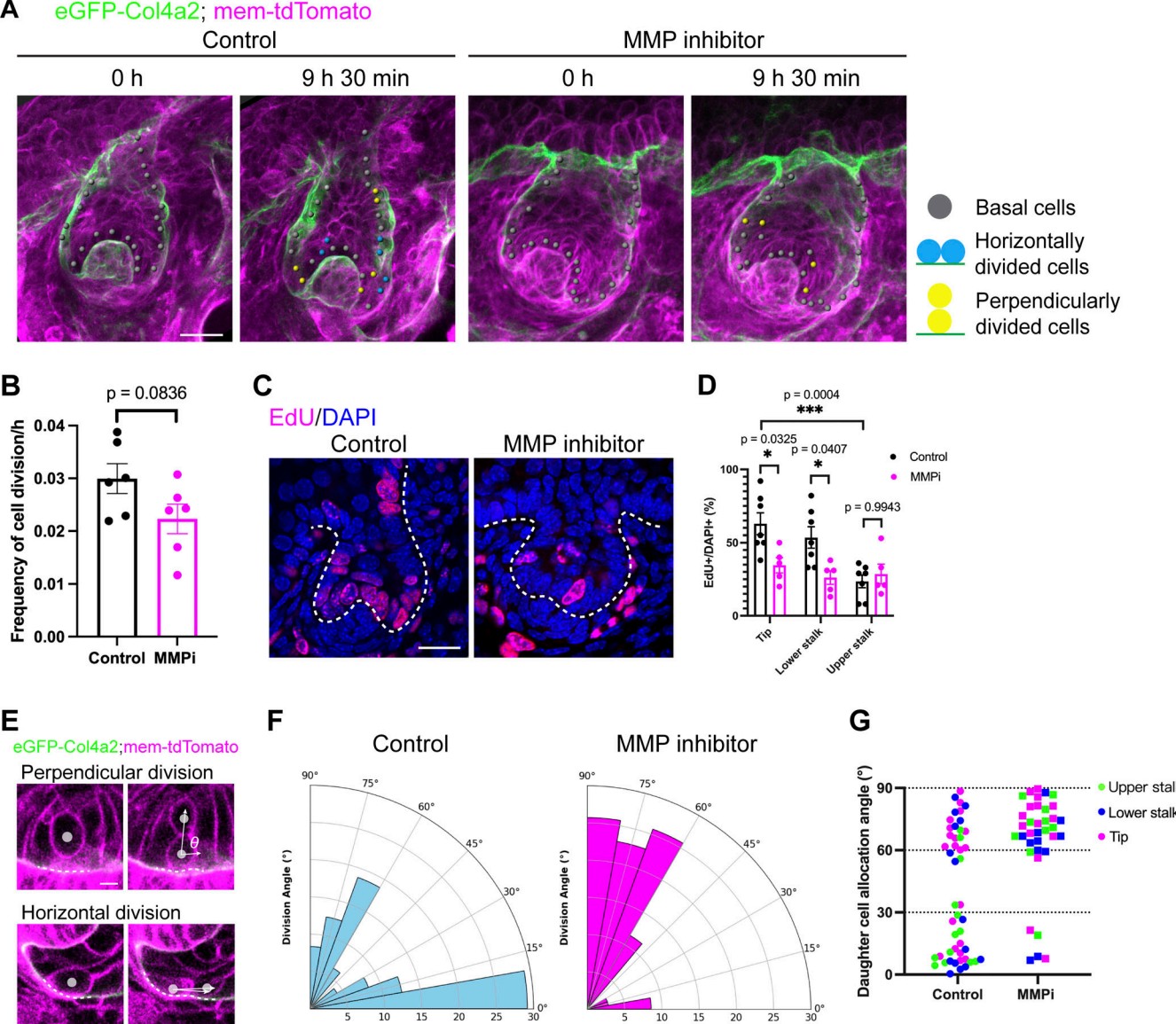

Figure 5. **Matrix metalloproteinase inhibition alters the angle of daughter cell allocation. (A)** Snapshot images and cell division tracking data from developing hair follicles in skin explants from *eGFP-Col4a2; mem-tdTomato* mouse embryos cultured under control or MMP-inhibited conditions. The BM and cell membrane were visualized with eGFP-COL4A2 (green) and mem-tdTomato (magenta), respectively. Scale bars: 20 μm. **(B)** Frequency of cell division in epithelial basal progenitors in the experiments in A (*n* = 6 follicles from five explants, each from an independent experiment for the control, *n* = 6 follicles, each from a separate explant in an independent experiment for the inhibitor-treated). A two-tailed unpaired *t* test was used. **(C)** EdU incorporation assays. Dividing cells were labeled with EdU (magenta). All the cells were counterstained with DAPI (blue). Scale bar: 20 μm. **(D)** Percentage of EdU-positive basal epithelial cells in the tip, lower stalk, and upper stalk regions (*n* = 7 follicles from three explants, each from an independent experiment for the control, and *n* = 5 follicles from three explants, each from an independent experiment for the inhibitor-treated). Values: means ± SD. Two-way ANOVA with Tukey's post-hoc tests was used. **(E)** Daughter cell allocation angle of proliferating basal epithelial progenitors relative to the BM zone in the experiments in A. Images of perpendicular and horizontal divisions are shown. Scale bar: 5 μm. **(F and G)** Pie graphs and a plot illustrating the angle of daughter cell allocation relative to the BM under control and MMP inhibitor treatment conditions (*n* = 48 cells from 11 follicles from seven explants, each from an independent experiment for the control, and *n* = 36 cells from eight follicles from eight explants, each from an independent experiment for the inhibitor-treated). All statistical tests in this figure assumed normal data distribution, but this was not formally tested.

observed dynamics (Fiore et al., 2020; Zamir et al., 2008). To address these issues, tagging endogenous ECM proteins with fluorescent markers is needed for more accurate analysis. Although several successful cases of tagged core BM proteins have been reported in *C. elegans* (Keeley et al., 2020; Matsuo et al., 2019; Naegeli et al., 2017) and *Drosophila* (Ramos-Lewis et al.,

2018), achieving this goal in mammals remains challenging. This is due not only to the complex assembly of the BM and its interactions with cell surface receptors and other BM molecules (Hynes, 2009; Timpl and Brown, 1996) but also to the lack of experimental systems that efficiently screen whether fluorescently tagged BM proteins are integrated into intact BMs

**Normal HF development**

**Figure 6. Model depicting the role of basement membrane dynamics in hair follicle morphogenesis.** This model illustrates the spatially distinct molecular and tissue-level dynamics of the BM, highlighting its pivotal role in regulating cell and tissue dynamics during hair follicle morphogenesis. The green lines indicate BMs. Magenta cells and BM zones are highlighted tracer epithelial basal cells and BM zones, respectively. The blue cells are epithelial progenitors that divide horizontally relative to the BM, and the red cells are epithelial progenitors that divide perpendicularly to the BM. In normal hair follicle development, the following key features are observed: (1) the rates of COL4A2 turnover and BM expansion exhibit spatial gradients, and these processes are coupled; (2) epithelial progenitors undergo both horizontal and perpendicular divisions; (3) basal epithelial cells move in unison with the directionally expanding underlying BM; and (4) the epithelial tube results in directional elongation. In MMP inhibitor-treated hair follicles: (1) COL4A2 turnover and BM expansion are restrained; (2) progenitors alter the angle of daughter cell allocation from horizontal to perpendicular; and (3) directional epithelial elongation is halted, resulting in a wider organ shape.

without impairing their functions. Recent advances, such as *Lamb1Dendra2* and *mTurq2-Col4a1* mice, have shown promise but still face challenges, including low or no homozygosity and tissue abnormalities (Jones et al., 2024; Morgner et al., 2023).

In this study, we inserted short linker-tagged eGFP or mKikGR near the N-terminus of the 7S domain of the endogenous *Col4a2* gene, avoiding the putative N-terminus intermolecular covalent cross-linking sites and the collagenous triple-helical domain of the 7S domain (Glanville et al., 1985; Matsubayashi et al., 2017; Morin et al., 2001). Our results suggest that although 7S domains are intermolecularly cross-linked and play essential roles in collagen IV network formation and stability, their N-terminus ends might have structural tolerance for the insertion of exogenous molecules. This site can be used to tag them with other fluorescent proteins, bioactive molecules, and biosensors, such as morphogens, pH sensors, and calcium sensors, in BMs in vivo.

The two mouse lines with tagged collagen IV exhibit distinct homozygous phenotypes: while *eGFP-Col4a2* shows normal development, *mTurq2-Col4a1* is embryonic lethal (Jones et al.,

2024). Several factors could account for this difference. The insertion sites of the fluorescent proteins vary, with *eGFP-Col4a2* having an insertion between $A_{28}$ and $Q_{29}$ of *Col4a2*, whereas *mTurq2-Col4a1* targets *Col4a1* between $K_{28}$ and $G_{29}$. Since COL4A1 and COL4A2 form heterotetramers in an $\alpha1\alpha1\alpha2$ configuration, the difference in the number of fluorescent proteins per heterotetramer may influence their function. Variations in the fluorescent proteins and linker designs (our line uses a $(GGS)_4$ linker on both sides) may affect the functionality of collagen IV. Although these factors likely contribute to the phenotypic divergence, the underlying mechanisms remain unclear. Further functional and structural investigations, as well as the development of an effective screening system, are needed to elucidate the causes and better understand the impact of fluorescent tagging on molecular functionality and animal development.

**Coordinated cellular and basement membrane dynamics during morphogenesis**

Embryonic development involves large-scale cell movement and tissue deformation. Traditionally, the dynamics of cell behavior

during this process have been attributed primarily to the coordinated interplay of active cell shape changes, cell migration, and cell proliferation, whereas the ECM has often been considered a passive, stable substrate for these cell-autonomous activities (Baena-López et al., 2005; Lemke and Nelson, 2021; Manning and Peifer, 2019; Matsubayashi, 2022). In mouse hair follicles, important morphogenetic cell behaviors, such as increased cell proliferation and directional epithelial cell movement toward the leading edge, occur during development and regeneration (Morita et al., 2021; Rompolas et al., 2012). However, our BM live imaging and quantitative analysis revealed that the BM exhibits spatial gradients of expansion rates, resulting in directional expansion along the axis of organ elongation. Moreover, epithelial progenitors proliferate and move with the directionally expanding BM rather than actively migrating on a stationary BM. When BM expansion is limited by MMP inhibitors, the progenitors alter the angle of daughter cell allocation from horizontal to perpendicular, halting directional organ elongation and resulting in a wider organ. These findings suggest that directional BM expansion contributes to the coordinated and directed expansion of tubular epithelia during morphogenesis by regulating the angle of daughter cell allocation and cell displacement.

BM movement has been observed in other systems. For example, in avian embryos, vortex-like movement of fibronectin-containing sub-epiblastic ECM (including the BM) has been reported (Zamir et al., 2008). In *Hydra*, the BM and fibrillar matrix move together with the epithelia toward the tips of the feet and tentacles (Aufschnaiter et al., 2011). During salivary gland branching morphogenesis, directional movement of the collagen IV network has been observed (Harunaga et al., 2014). Interestingly, in this context, the collagen IV network moves from the tips to the stems of the branches, the opposite of the direction of BM movement in developing hair follicles. These examples emphasize the importance of quantitative measurement of composite tissue motion, encompassing both cellular and matrix components, to better understand how BM expansion actively contributes to organ growth.

**Mechanisms of basement membrane expansion**
The phenomenon of BM expansion raises important questions about its underlying mechanisms. While MMP activity contributes to BM turnover and homeostatic expansion, it cannot fully account for how expansion is driven. This is because simply removing and replacing BM components would likely result in homeostatic turnover without leading to significant expansion. The mechanical tension generated by cells and the ECM is likely central to BM expansion and directional organ growth. As epithelial cells proliferate, they can potentially generate directional tension that stretches the BM anisotropically, creating space to incorporate more free collagen IV molecules than the number of degraded molecules, leading to expansion along the main axis of the growing tissue (Chang et al., 2024; Khalilgharibi and Mao, 2021). Consistently, our findings show that BM lengthening occurs alongside epithelial proliferation with horizontal daughter cell allocation. However, when an MMP inhibitor is applied, the orientation of cell allocation shifts

from parallel to perpendicular relative to the BM, while cell proliferation continues. This suggests that the tension generated by parallel cell allocation, which is normally aligned with the expanding tissue (Box et al., 2019; Dekoninck et al., 2020; Gudipaty et al., 2017; Nestor-Bergmann et al., 2019), is adaptive to the surrounding ECM properties and dynamics. Thus, this proliferation-induced tension alone appears insufficient to drive BM expansion without MMP-dependent matrix remodeling.

The BM may also actively expand or shrink through the polymerization and crosslinking of its components (Diaz-de-la-Loza and Stramer, 2024; Serna-Morales et al., 2023). In our study, we observed that BM expansion is accompanied by increased polymerization of type IV collagens, indicating that this process is not merely a passive stretching but involves coordinated remodeling of the molecular structure of the BM. Further spatially and temporally detailed analysis of the cell behavior and BM dynamics, such as examining whether BM expansion proceeds locally before parallel cell divisions take place or vice versa, along with targeted perturbations of cellular and ECM activities, will be essential to fully understand the role of BM assembly in regulating cell behavior. Additionally, compositional changes in the BM could also facilitate BM and tissue expansion. For example, the incorporation of different BM components, such as laminins or perlecan, that alter stiffness or elasticity might allow for expansion (Pastor-Pareja and Xu, 2011; Töpfer et al., 2022).

Finally, tissue-scale BM motion (including expansion) can displace adhering cells, further contributing to the coordinated movement of both cells and the BM along a specific axis (Loganathan et al., 2016). Our findings of BM-coupled cell movements in developing mouse skin add further support to the idea that BM expansion is closely integrated with tissue-level mechanical forces and cellular dynamics.

**The role of MMPs in BM dynamics and morphogenesis**
Our results indicate that the spatiotemporal regulation of MMP activity is crucial for coordinating BM remodeling and epithelial morphogenesis. However, the mechanisms regulating MMP activity in hair follicles are largely unexplored. In contrast, studies in the mammary gland have demonstrated that MMP2-null glands exhibit defective terminal end bud invagination with excessive secondary branching, whereas MMP3-null glands show normal invagination but deficient secondary branching (Wiseman et al., 2003). These findings suggest that the regulation of MMP activity must be finely tuned to achieve proper tissue morphogenesis. Understanding how MMPs control BM dynamics in the development of hair follicles and other organs will be critical for elucidating the mechanisms underlying tissue-specific morphogenesis.

In summary, the BM live-imaging approach established in this study has the potential for studying the role of BM dynamics in a wide range of biological phenomena in mammals, including early development, organogenesis, homeostasis, regeneration, and disease progression under conditions such as cancer. Therefore, this imaging technique makes a significant contribution to our quest for a deeper understanding of the functional importance of BM dynamics in multicellular systems.

## Materials and methods

### Mice

All mouse experimental procedures in this study were approved by the Institutional Animal Care and Use Committee of the RIKEN Kobe Branch. The care and handling of the mice adhered to the ethical guidelines of the RIKEN Kobe Branch as well. To visualize BM dynamics, we generated *eGFP-Col4a2* knock-in mice (accession no. CDB0100E: https://large.riken.jp/distribution/mutant-list.html). A 717-bp eGFP sequence (without a stop codon) was inserted into the three amino acids after the end of the predicted signal peptide ($M_1$–$G_{25}$) in exon 3 of *Col4a2*. To minimize the structural interference of the inserted fluorescent proteins with COL4A2 functions, a *Gly-* and *Ser*-rich flexible linker, $(GGS)_4$, was added to both ends of the eGFP gene. To construct a donor vector that contained homologous arms (319 and 286 bp), the *Col4a2* genomic sequence around the eGFP insertion site was cloned and inserted into pBluescript II. The cDNA sequences of eGFP and the linkers were integrated into the insertion site within the donor vector. C57BL/6 zygotes were subjected to microinjection with a mixture of Cas9 protein (100 ng/μl), crRNA (50 ng/μl)/tracrRNA (100 ng/μl) and the donor vector (10 ng/μl), and then the zygotes were transferred to pseudopregnant females to obtain founder knock-in mice (Abe et al., 2020). The crRNA (5′-ACUCCUGGCUCAGAGCGUCUGUUUUAGAGCUAUGCUGUUUUG-3′) and the tracrRNA (5′-AAACAGCAUAGCAAGUUAAAAUAAGGCUAGUCCGUUAUCAACUUGAAAAAGUGGCACCGAGUCGGUGCU-3′) were purchased from FASMAC. The knock-in mice were screened via PCR-based genotyping using the three primer pairs (LF+LR, RF+RR, and LF+RR) shown in Fig. S1. The primers used were LF (5′-GCTGCTGCTAGCAACTGTGACAG-3′), LR (5′-GTGCAGATGAACTTCAGGGTCAGC-3′), RF (5′-GGTCCTGCTGGAGTTCGTGACC-3′), and RR (5′-AGCACACACCTACAATGCACACG-3′). The LF and RR primer pair was designed on the *Col4a2* allele and the RF and LR pair was designed on the *eGFP* sequence. Three primer pairs were used for PCR genotyping to distinguish the WT allele (LF-RR; 182 bp) and the *eGFP-Col4a2* knock-in allele (LF-LR; 222 bp, RF-RR; 242 bp; LF-RR; 971 bp) (Fig. S1 B). We used genome sequencing to confirm that the linker and *eGFP* construct were correctly inserted into the *Col4a2* gene (Fig. S1 C). To investigate BM turnover, *mKikGR-Col4a2* mice (accession no. CDB0126E: https://large.riken.jp/distribution/mutant-list.html) were generated via the same strategy. The plasmid for mKikGR was purchased from Addgene (#54656) (Habuchi et al., 2008). The knock-in mice were screened via PCR-based genotyping using the three primer pairs (LF1+LR1, RF1+RR, LF1+RR). The primers used were LF1 (5′-AAGACTGGGATCATGGACCG-3′), LR1 (5′-TGGCTCCCATTTGACGGTCTTCC-3′), RF1 (5′-ACAGTCATAGAGGGCGGACCTCT-3′), and RR (5′-AGCACACACCTACAATGCACACG-3′). The LF1 and RR primer pair was designed on the *Col4a2* allele and the RF1 and LR1 pair was designed on the *mKikGR* sequence. Three primer pairs were used for PCR genotyping to distinguish the WT allele (LF1-RR; 363 bp) and the *mKikGR-Col4a2* knock-in allele (LF1-LR1, 689 bp; RF1-RR, 751 bp; and LF1-RR, 1,134 bp). To visualize cells and the BM simultaneously, *eGFP-Col4a2* mice were bred with *mTmG* mice (JAX Strain no. 007576; The Jackson Laboratories). All the mice except for *mKikGR-Col4a2* were bred with FVB/NJcl

mice (CLEA Japan) to avoid imaging interference from melanin deposition.

### Immunohistochemistry

For fluorescence immunohistochemistry, the tissues were embedded in an optimal cutting temperature (OCT) compound, frozen, and cryosectioned (10–16 μm). The sections were fixed with 4% paraformaldehyde (PFA) in phosphate-buffered saline (PBS) for 5 min at 4°C, blocked and permeabilized with blocking buffer (0.5% skim milk/0.25% fish skin gelatine/0.5% Triton X-100/PBS) for 1 h at room temperature (RT), and then incubated with the primary antibodies overnight at 4°C. The sections were subsequently washed with PBS and incubated with fluorescence-labeled secondary antibodies for 2 h at RT. The sections were then washed with PBS and mounted with DAKO Fluorescent Mounting Medium. Fluorescent signals were captured under a TCS SP8X (Leica) confocal microscope with LAS X software (version [BETA] 3.5.7.23723).

For whole-mount immunostaining, the tissues were fixed with 4% PFA in PBS for 5–60 min at 4°C. The tissues were blocked and permeabilized with blocking buffer (0.5% skim milk/0.25% fish skin gelatine/0.5% Triton X-100/PBS) for 1 h at 4°C and then incubated with the primary antibodies overnight at 4°C. The sections were subsequently washed with PBT (0.2% Tween-20 in PBS, pH 7.4) and incubated with fluorescence-labeled secondary antibodies for 2 h at RT. The sections were then washed three times with PBT (0.2% Tween-20 in PBS, pH 7.4) for 30 min and dehydrated in 50% methanol/PBS for 10 min, 100% methanol for 5 min, 100% methanol for 10 min, 50% benzyl alcohol with benzyl benzoate (1:2) (BABB)/methanol for 5 min, 100% BABB for 5 min, and 100% BABB for 10 min. Fluorescent signals were captured under a TCS SP8X (Leica) confocal microscope with LAS X software (version [BETA] 3.5.7.23723). The antibodies used in this study are shown in Table S1.

### Ex vivo culture of embryonic dorsal skin

The ex vivo culture of embryonic dorsal skin was conducted as described previously (Morita et al., 2021). The pregnant mice were euthanized by cervical dislocation and embryos at embryonic day (E) 12.5–13.5 were collected. The entire dorsal skin tissues of the embryos were dissected with 25G needles under a stereo microscope, and a piece approximately one-fourth the size of the entire skin was excised for culturing. Approximately, 10 μl of collagen type I-A (Nitta Gelatin) gel solution was added to an empty 35-mm Lumox dish (Sarstedt) on ice, and then the skin piece was embedded into the collagen gel with the dermis facing upward. The skin/collagen gel drop was incubated in a humidified $CO_2$ incubator with 5% $CO_2$ at 37°C for 30 min to set the collagen gel, and it was subsequently submerged in 1 ml of DMEM/Ham's F12 with L-glutamine and sodium pyruvate (Wako) supplemented with 20% fetal bovine serum (GIBCO), 100 U/ml of penicillin, and 100 μg/ml of streptomycin (GIBCO), 1X GlutaMAX (GIBCO), 10 mM HEPES (GIBCO), and 100 μg/ml ascorbic acid (Sigma-Aldrich). The explants were cultured at 37°C in a humidified atmosphere with 5% $CO_2$.

To inhibit MMP enzymatic activity, on day 3 of the culture, the culture medium was replaced with the same medium containing 3 µM of batimastat (ab142087; Abcam).

## EdU incorporation assays

On Day 4 of the culture, 10 µM of EdU was introduced into the culture medium of the skin explants using a 10 mM stock solution from the Click-iT Plus EdU Cell Proliferation Kit (C10640; Thermo Fisher Scientific). For the MMP inhibitor-treated group, EdU was added 16 h after the addition of batimastat. After 2 h (for Fig. 2 F) or 3 h (for Fig. 5 C) of incubation with the EdU solution, the skin explants were fixed and permeabilized according to the manufacturer's protocol. EdU was detected via a Click-iT Plus reaction, followed by Hoechst 33342 or DAPI nuclear counterstaining. 30-µm-thick 3D images of hair follicles were used to count the number of EdU-positive nuclei. To quantify the number of EdU-positive nuclei, a freehand line was drawn on the BM (eGFP-COL4A2 signal) using Imaris (version 8.4.2; Bitplane). The tip region was defined as the area between the two lowest bending points (indicated by the blue line in Fig. 2 F). From the interfollicular junction to the lowest protrusion point, a line was drawn, and the midpoint was determined. The segment from the bending point to the midpoint was defined as the lower stalk region (indicated by the orange line in Fig. 2 F), whereas the segment from the interfollicular junction to the midpoint was defined as the upper stalk region (indicated by the gray line in Fig. 2 F).

## Live imaging of embryonic dorsal hair follicles

After skin tissue dissection, explants were cultured in a $CO_2$ incubator for 3 days to induce hair follicle formation. Live imaging of dorsal hair follicles was performed under a TCS SP8X (Leica) confocal microscope with LAS X software (version [BETA] 3.5.7.23723), a stage top incubator (Tokai Hit), and a 25× water-immersion objective lens (HC FLUOTAR 25×/0.95 W VISIR; Leica) or an LSM980 (Carl Zeiss) with a two-photon laser unit Chameleon Discovery NX (Coherent), Zen software (version 2.3), a stage top incubator, and a 32× multi-immersion objective lens (C-Achroplan 32×/0.85 W Corr M27; Zeiss). For two-photon laser imaging, a 920-nm laser and a 1,040-nm laser were used to excite eGFP and tdTomato, respectively. For long-term live imaging, immersion water was dispensed onto the objective lens using a Water Immersion Micro Dispenser (Leica) or Liquid Dispenser (Merzhauser Wetzlar). Successive optical section images were obtained as stacks at 512 × 512 pixels for the x–y plane and a z-stack step size of 1–1.5 µm, covering a total tissue depth of ∼30 µm. These image stacks were acquired at intervals of 10–30 min.

## Fluorescence recovery after photobleaching (FRAP)

Photobleaching was performed under a TCS SP8X (Leica) confocal microscope or an LSM980 (Carl Zeiss) with the same accessory settings described above. A 488-nm laser (100% power) was used to induce photobleaching of eGFP-COL4A2 in an ROI with a length of 15 ± 2 µm and a width of 2–3 µm. The fluorescence intensity values of the BM region were acquired using ImageJ (version 1.53). First, a line was drawn on the BM of the

prebleached and bleached regions on three single z-plane images, which included the target z-stack plane and its upper and lower stacks, and fluorescence intensity values were obtained for each pixel. To create line graphs, the fluorescence intensity values of each line from the three z-planes were averaged, and then they were divided by the average fluorescence intensity values of the prebleached times. The unique protein dynamics of COL4A2, including its very limited diffusibility and slow turnover rates within the BMs, especially at the lower stalk and upper stalk regions, necessitate prolonged live-imaging sessions to capture the full recovery process. Such prolonged observation periods are required to accurately capture the plateau of the eGFP-COL4A2 recovery intensity. However, tracking the photobleached BM zone throughout the recovery period has become increasingly challenging because of the morphological changes in embryonic skin tissues, including the BM itself. The potential increase and decrease in the total amount of eGFP-COL4A2 within the BM during this extended recovery process underscore the non-static nature of our experimental system. As a result, due to the difficulty in reaching a clear plateau in our FRAP data, we used simple linear regression to fit the fluorescence recovery data, providing a more consistent analysis under these conditions.

## Photoconversion

Tissues expressing mKikGR-COL4A2 were imaged with a 488-nm laser under a TCS SP8X (Leica) confocal microscope with the same accessory settings described above. A 405-nm diode laser (20% power with five iterations) was used for photoconversion on rectangular ROIs (15 × 15 µm). The photoconverted mKikGR was imaged with a 552-nm laser.

## Measurement of changes in basement membrane length and cell position

To ensure accurate and consistent measurement of the BM length over time, we carefully selected and monitored the appropriate z-plane at each time point. We have provided detailed visual explanations in Fig. S2.

Next, to establish reference points for BM length measurement, photobleaching was performed on the BM. The photobleaching sessions utilized a 488-nm laser set at 100% power, with three sets of 20 iterations each. The ROI was photobleached across 10 ± 3 continuous z-planes (with 1-µm intervals), centered on the target z-plane, to cover the z-stacks where BM regions appear as clear line-like structures on different z-planes, leading to the formation of a 3D cuboidal bleached region. As the eGFP-COL4A2 signal recovers over time, identifying the reference points (the edges of distinct photobleached edges) in regions where the turnover time of eGFP-COL4A2 is short becomes increasingly difficult. Therefore, at intervals of ∼2–3 h of culture following each photobleaching session, the edges of the photobleached areas were identified, and a new ROI was

drawn along these edges. The ROI size (x-y axis) and positions were adjusted for each photobleaching session according to the position of the border between photobleached and non-photobleached BM regions. This procedure ensures that the ROI is consistently located near the originally identified BM region, allowing accurate measurement of BM length at different time points corresponding to the same z-axis. After 7 h, the z-planes where the clear BM of each compartment could be observed were selected for BM length measurement, as shown in Fig. S2 B. As no significant morphological changes were observed at the boundaries of the 3D cuboidal bleached area during the imaging period, we can conclude that the BM maintains a consistent shape and expansion rate, even with shifts in the circumferential location of the hair follicle (Fig. S2 C). This suggests that any potential displacement around the circumference does not affect the accuracy of the measurements.

To analyze changes in BM length and cell position simultaneously, *eGFP-Col4a2* mice were crossed with *mTmG* mice (JAX stock #007576). To make reference points for measurement, photobleaching was performed on the BM as described above. An imaging interval of 20–35 min for 3D live imaging enabled the consistent identification of the same cells (see details in Fig. S4). The length of the BM was measured between the bleached edges of eGFP-COL4A2 in the upper stalk and lower stalk regions. The cell positions were determined by measuring from the upper bleached edge of the reference point on the BM to the midpoint of the cell membrane of the target cell in the upper stalk and lower stalk regions.

### Measurement of the thickness of the basement membrane

Single z-plane confocal or two-photon microscope images of the eGFP-COL4A2 signal were used as proxies for the BM structure. To measure the thickness, the perimeter of the eGFP-COL4A2 signal was enclosed by drawing a line using the freehand selection tool in ImageJ2 (version 2.14), creating an area with a long-axis length of 5–10 μm. This area was generated on the BM in the tip, lower stalk, and upper stalk regions (see Fig. 2 D). The average BM thickness was calculated by dividing the area by the long-axis length for each region.

### Measurement of the angle of epithelial basal cell division

The orientation of epithelial basal cell division was determined by measuring the angle between a line connecting the centers of two daughter cells and the reference line drawn parallel to the underlying BM (Lough et al., 2019). The plasma membrane and BM were simultaneously visualized with mem-tdTomato and eGFP-COL4A2, respectively. The cells were tracked manually through identification of the cell shape and position, referring to mem-tdTomato at each time point using Imaris (version 8.4.2; Bitplane). The cells that underwent division were identified by observing the following sequence of events: an initial increase in cell volume, mitotic cell rounding, cleavage furrow formation, and, finally, the separation of the daughter cells.

### Detection and quantification of mRNA expression in tissues

To detect single RNA molecules in mouse embryonic tissues, a whole-mount RNAscope was employed. Briefly, freshly dissected tissues were embedded in the collagen gel and incubated in a humidified $CO_2$ incubator with 5% $CO_2$ at 37°C. On day 3 of the culture (16 h before fixation), 3 μM batimastat was added to the cultures of the MMPi group. On day 4 of the culture, the skin explants were fixed in 4% PFA/PBS. All the samples were incubated with hydrogen peroxide for 1 h, washed three times with PBT (0.1% [vol/vol] Tween-20 in PBS, pH 7.4) and hybridization was carried out overnight at 40°C. After hybridization, the expression patterns of the target mRNAs were detected and visualized according to the manufacturer's protocol for the RNAscope Multiplex Fluorescent Reagent Kit v2 (322381; ACD). The stained tissues were washed with PBS, stored in Ultra-mount Permanent Mounting Medium (DAKO), and photographed using a TCS SP8X (Leica) confocal microscope. For signal quantification, an enclosed area was drawn using the freehand selection tool of ImageJ2 (version 2.14) to define the "ROI area" on the hair follicle at the basal layer, dermal papilla, dermal sheath, suprabasal epithelial layer, and mesenchymal regions. *Col4a2*-positive signal particles were manually counted.

### Statistical analysis and reproducibility

No statistical method was used to predetermine the sample size. Statistical parameters, including the numbers of samples and replicates, types of statistical analysis, and statistical significance are indicated in the results, figures, and figure legends. For the quantitative analysis, we used five or more biological replicates for each experiment. Statistically significant differences are indicated in each figure (*$P < 0.05$, **$P < 0.01$, ***$P < 0.001$, and ****$P < 0.0001$). Specific statistical tests used in each experiment are described below.

Fig. 2, C, E, and G: One-way ANOVA with Tukey's post-hoc tests.
Fig. 2, I and J: Two-tailed unpaired *t* tests.
Fig. 3 C: One-way ANOVA with Tukey's post-hoc tests.
Fig. 4, D, F, N, O, S, and U: Two-tailed unpaired *t* tests.
Fig. 4, E, J, and L: Two-way ANOVA with Tukey's post-hoc tests.
Fig. 5 B: Two-tailed unpaired *t* test.
Fig. 5 D: Two-way ANOVA with Tukey's post-hoc tests.
Fig. S6 B: Two-tailed unpaired *t* tests.

### Online supplemental material

Fig. S1 shows the generation of eGFP-Col4a2 knock-in mice. Fig. S2 shows procedures for re-photobleaching of BMs. Fig. S3 shows an analysis of 3D BM structure in hair follicles. Fig. S4 shows time-lapse snapshots of BM and cell tracking in hair follicles. Fig. S5 shows imaging of mKikGR-COL4A2 in developing hair follicles. Fig. S6 shows Col4a2 mRNA expression in control and MMP inhibitor-treated hair follicles. Video 1 shows a 3D time-lapse video of eGFP-COL4A2 signals in embryonic dorsal skin explants. Video 2 shows FRAP experiments with eGFP-COL4A2 in hair follicles. Video 3 shows pulse-chase video in mKikGR-Col4a2 follicles, showing BM turnover dynamics. Video 4 shows FRAP experiments under MMP inhibition. Video 5 shows the time-lapse of epithelial progenitor cell division angles. Table S1 lists key resources used in this study. Data S1 includes raw data used for quantification in this study.

**Data availability**

All image data in this study have been stored in the SSBD repository at https://doi.org/10.24631/ssbd.repos.2023.04.298. All data that support the findings of this study are available within the paper and its supplementary information files.

## Acknowledgments

We thank the members of the Fujiwara Laboratory for valuable reagents and discussions; the Laboratory for Animal Resources and Genetic Engineering for their technical assistance; Shigeru Kuratani (RIKEN) for sharing animal resources; Yasuko Tomono (Shigei Medical Research Institute) for antibodies against the Collagen IV α2 chain; Yasunori Miyamoto (Ochanomizu University) for his generous support on K. Hashimoto's research in the Fujiwara Laboratory; and Shinji Deguchi (Osaka University) for his advice on FRAP analysis.

This work was funded by RIKEN intramural funding, the RIKEN BDR Stage Transition Project, the JSPS Kakenhi (19K22631, 20H03706, 22K19453), the MEXT Kakenhi (23H04927, 23H04928), and the JST CREST (JPMJCR1926) (all to H. Fujiwara). D. Wuergezhen was supported by the RIKEN International Program Associate (IPA) and the Interdisciplinary Program for Biomedical Sciences, Osaka University. K. Hashimoto was supported by the Program for Leading Graduate School of Ochanomizu University.

Author contributions: D. Wuergezhen: Conceptualization, Data curation, Formal analysis, Funding acquisition, Investigation, Methodology, Project administration, Resources, Software, Supervision, Validation, Visualization, Writing - original draft, Writing - review & editing, E. Gindroz: Conceptualization, Investigation, Methodology, R. Morita: Methodology, Writing - review & editing, K. Hashimoto: Investigation, Writing - review & editing, T. Abe: Resources, H. Kiyonari: Resources, H. Fujiwara: Conceptualization, Data curation, Funding acquisition, Methodology, Project administration, Resources, Supervision, Validation, Writing - review & editing.

Disclosures: D. Wuergezhen reported having a patent pending for imaging techniques of the basement membrane. E. Gindroz reported having a panent pending for imaging techniques of the basement membrane. H. Fujiwara reported having a patent pending for imaging techniques of the basement membrane.

Submitted: 1 April 2024

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

# Supplemental material

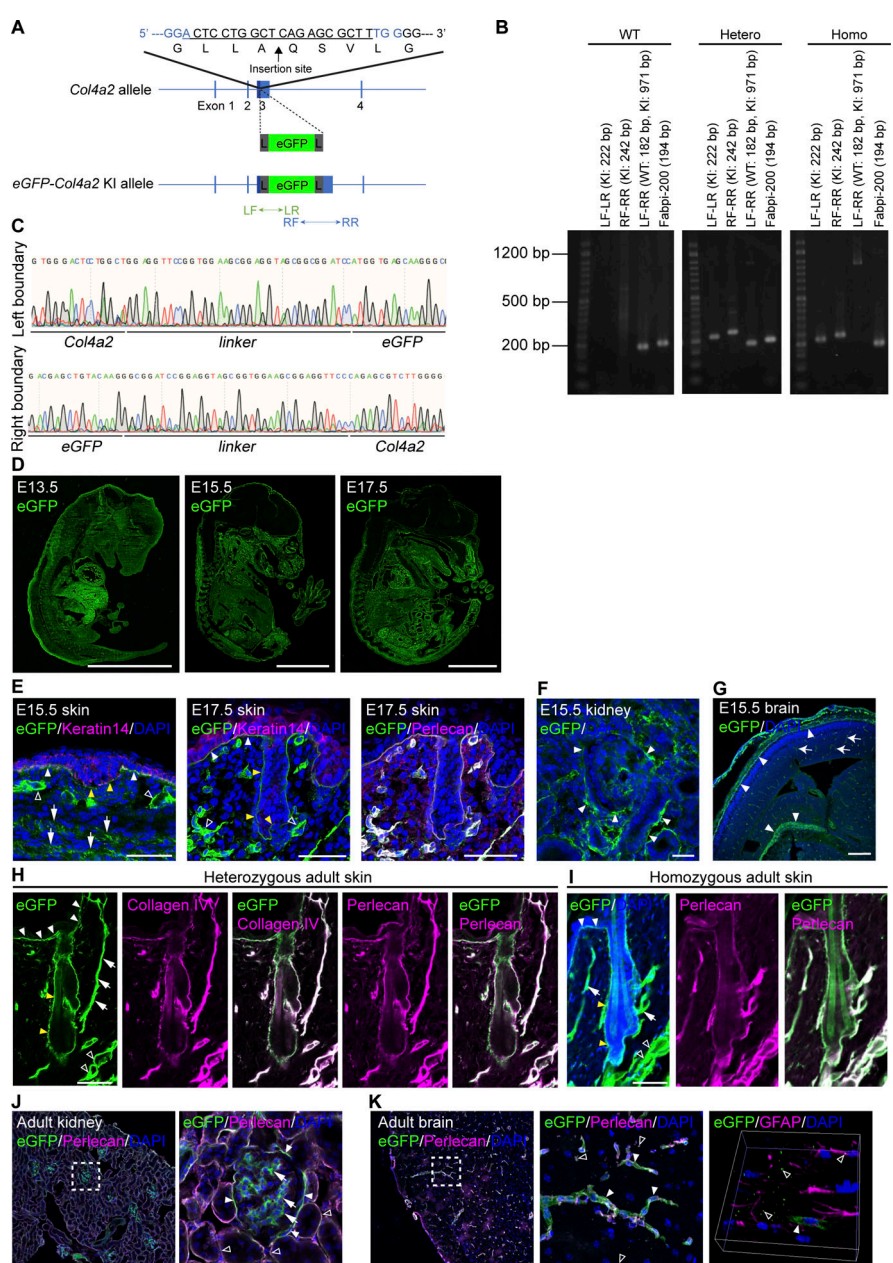

Figure S1. **Generation of *eGFP-Col4a2* knock-in mice. (A)** Targeting strategy for generating *eGFP-Col4a2* knock-in mice. An eGFP protein was inserted into the N-terminus region of the 7S domain of the COL4A2 protein (between $A_{28}$ and $Q_{29}$). To minimize the structural interference of the inserted fluorescent protein with COL4A2 protein structure and functions, a glycine- and serine-rich flexible linker, GGSGGSGGSGGS, was added to both ends of the EGFP protein. L, linker; LF, left forward primer; LR, left reverse primer; RF, right forward primer; RR, right reverse primer. **(B)** Image of PCR screening used to distinguish wild-type, heterozygous, and homozygous *eGFP-Col4a2* mice. Fabpi-200: An internal control primer pair for detecting the *Fabpi* gene. A 50 bp DNA ladder RTU (GeneDirex) was used. **(C)** Genome sequences of boundaries between the *Col4a2* gene and linkers. **(D)** Immunofluorescence images of E13.5, E15.5, and E17.5 embryonic tissues from *eGFP-Col4a2* mice stained for eGFP. Scale bars: 3 mm. **(E–G)** Immunofluorescence images of tissues from *eGFP-Col4a2* mouse embryos. **(E)** At E15.5 (left panel), the eGFP (green) signal is detected at the epidermal BM (white closed arrowheads) below the keratin 14-positive (magenta) basal epidermis, the BMs of the hair placode (yellow closed arrowheads), the dermal vessel-like structures (open arrowheads), and the panniculus carnosus muscle-like layers (arrows). At E17.5 (middle and right panels), eGFP (green) is detected in perlecan-positive (magenta) epidermal and vessel-like BMs. **(F)** In the E15.5 kidney, eGFP (green) appeared around the tubular epithelial structures (closed arrowheads). **(G)** In the E15.5 brain, eGFP (green) appears in pial BM zones (closed arrowheads) and vessel capillaries (arrows). DAPI (blue) was used for nuclear counterstaining. Scale bars: (E and F) 20 μm, (G) 200 μm. **(H–K)** Immunofluorescence images of adult tissues from *eGFP-Col4a2* mice. **(H)** In P56 adult telogen dorsal skin tissues from het mice, eGFP (green) colocalizes with collagen IV (magenta) and perlecan (magenta) in the BMs of the epidermis (closed white arrowheads), hair follicle epithelium (closed yellow arrowheads), arrector pili muscle (arrows), and vessel-like tissues (open arrowheads). **(I)** In P56 adult telogen dorsal skin tissues from homo mice, eGFP (green) is distributed as it is in het mice and colocalizes with perlecan (magenta). **(J)** In adult kidneys, eGFP (green) is detected in Bouman's capsules (closed arrowheads), the mesangial matrix (arrows), and collecting ducts (open arrowheads). The same field of view is presented in Fig. 1 F. **(K)** In the subventricular zone, eGFP (green) colocalizes with perlecan (magenta) in capillaries (closed arrowheads) and fractone-like structures (open arrowheads) around GFAP+ (magenta) astrocytes. DAPI (blue) was used for nuclear counterstaining. Scale bars: 50 μm. Source data are available for this figure: SourceData FS1.

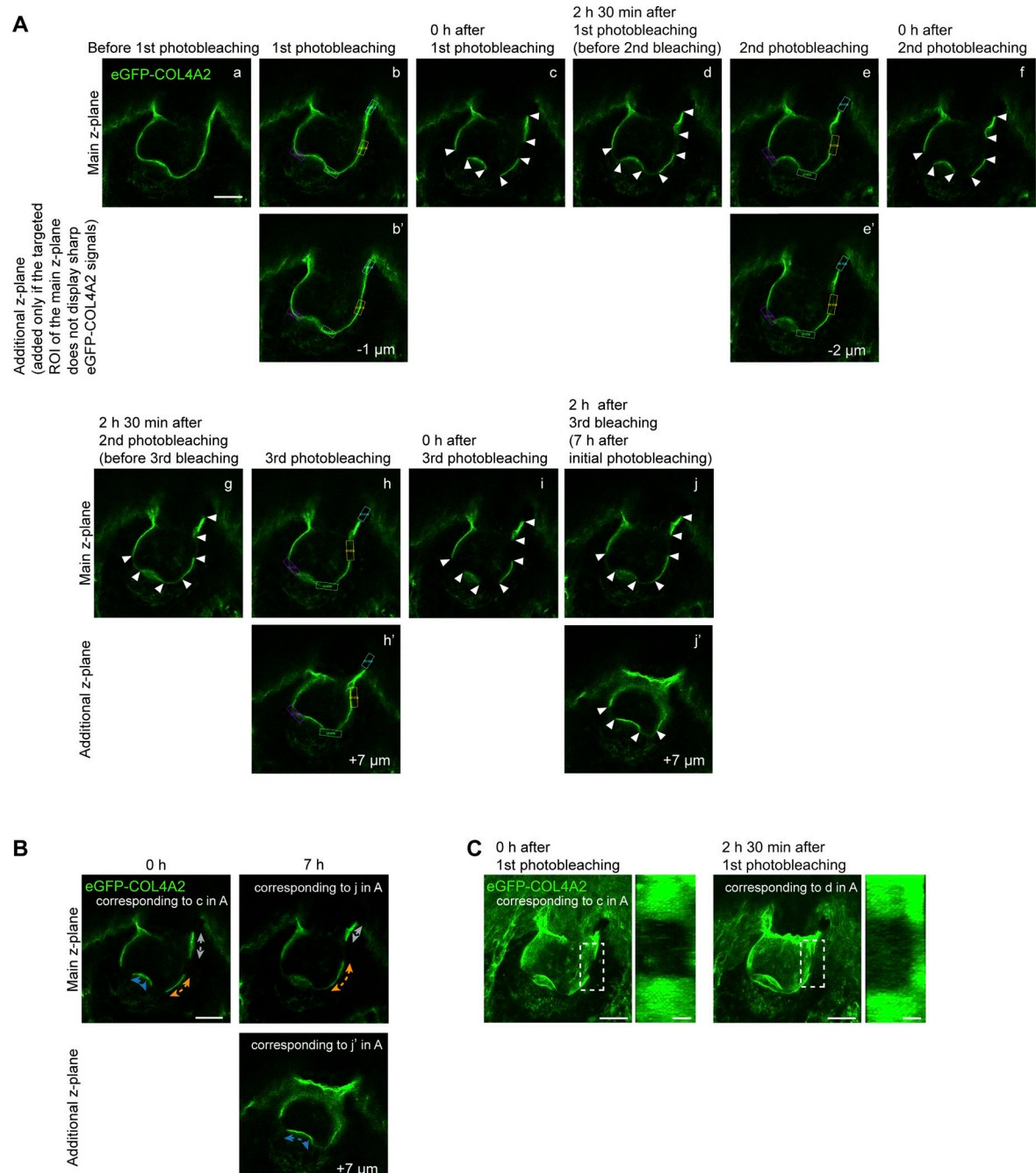

Figure S2. **Re-photobleaching procedures for the measurement of basement membrane expansion. (A)** Images of re-photobleaching procedures (see Materials and methods for detailed procedures and explanations). A z-plane exhibiting sharp and narrow signals from the BM within the targeted area was selected for photobleaching (panels a, b, and b'). The ROIs (indicated by rectangular boxes) were set on the targeted z-plane. The ROI was photobleached across 10 ± 3 continuous z-planes (with 1 μm intervals), centered on the target z-plane. For example, the BMs in the tip and lower stalk regions presented sharp and narrow signals; thus, the purple, green, and yellow ROIs were set around the main z-plane (panel b). As the upper stalk BM showed sharper signals in the 1 μm lower z-plane (additional z-plane: panel b'), the cyan ROI was set around the additional z-plane. Additional z-planes are displayed only if some target areas in the main z-plane do not show sharp eGFP-COL4A2 signals. The same photobleaching procedure was applied to each ROI. At intervals of 2–2 h and 30 min of culture following each photobleaching session, the edges of the photobleached areas were identified (distinguishable edges are indicated by arrowheads: panels c, d, f, g, i, j, and j'), and a new ROI was drawn along the identified photobleached edges (panels e, e', h, and h'). This procedure was repeated twice until the end of the imaging session (7 h after initial photobleaching: panels j and j'). Scale bar: 20 μm. **(B)** Images of BM length changes. These images show the procedures used to quantify BM length changes in A. Measurements were conducted in different z-planes to determine the length of the BM at the upper and lower stalk regions (upper panels) and the tip region (upper and lower panels). Scale bar: 20 μm. **(C)** Maximum projection images of eGFP-COL4A2 in hair follicles in A are presented. Side views of the dotted cuboidal areas are shown. As no significant shape changes were observed at the boundaries of the 3D cuboidal bleached area during the imaging period, we can conclude that the BM maintains a consistent shape and expansion rate, even with shifts in the circumferential location of the hair follicle. This suggests that any potential displacement around the circumference does not significantly affect the accuracy of the measurements. Scale bars: 20 μm for maximum projection images; 5 μm for magnified side view images.

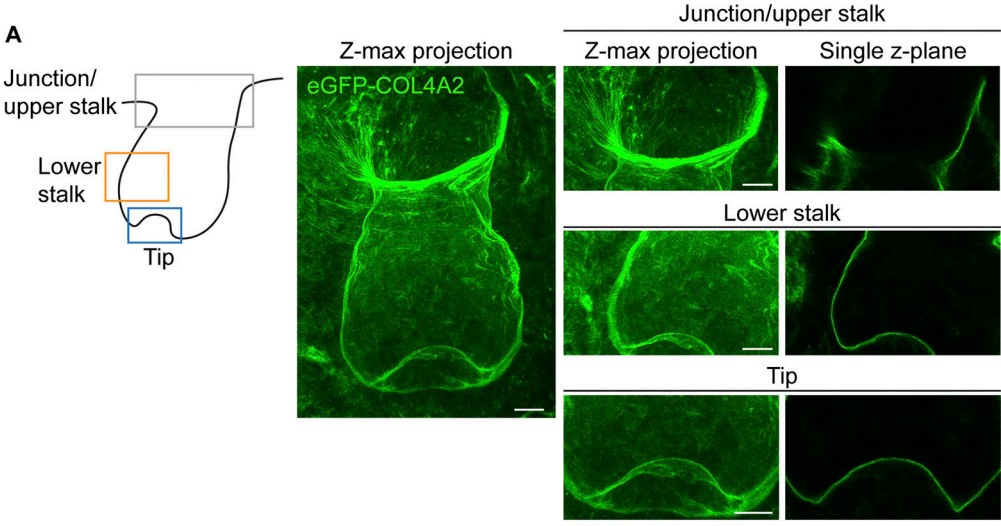

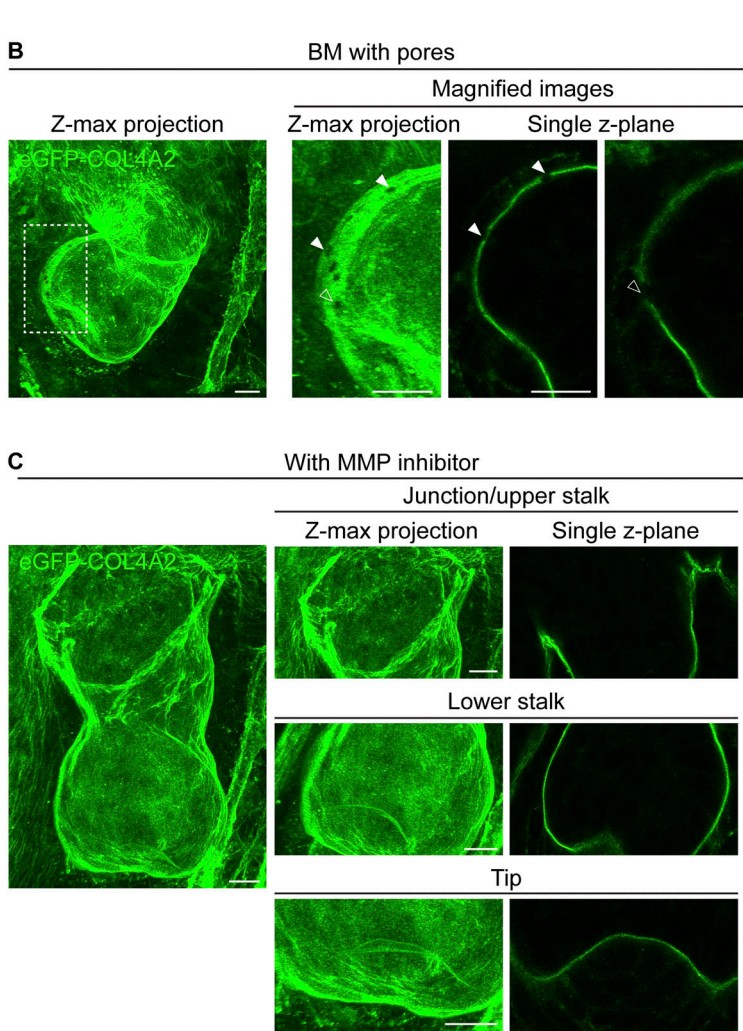

Figure S3. **Analysis of the 3D basement membrane structure. (A)** Confocal images of the hair follicles enclosed by a continuous line-like BM structure. Images show z-stack maximum projection images and single z-plane images of developing hair follicles with a continuous line-like BM structure at the tip (observed in 9 out of 13 follicles examined), lower stalk (observed in 12 out of 13 follicles examined), and upper stalk regions (observed in all 13 follicles examined). These hair follicles were developed in cultured skin explants from E12.5 embryos of *eGFP-Col4a2* mice ($n$ = 13 follicles from five explants, each from an independent experiment). Scale bars: 10 μm. **(B)** Confocal z-stack maximum projection and single z-plane images of the hair follicles exhibiting pores (arrowheads) in its lower part. The BM with pores (a dotted rectangular area) is magnified. Scale bars: 10 μm. **(C)** Confocal z-stack maximum projection and single z-plane images of the hair follicles treated with an MMP inhibitor (batimastat). No clear pore structures were observed in the 13 MMP inhibitor-treated hair follicles examined. ($n$ = 13 follicles from five explants, each from an independent experiment for the inhibitor-treated condition). Scale bars: 10 μm.

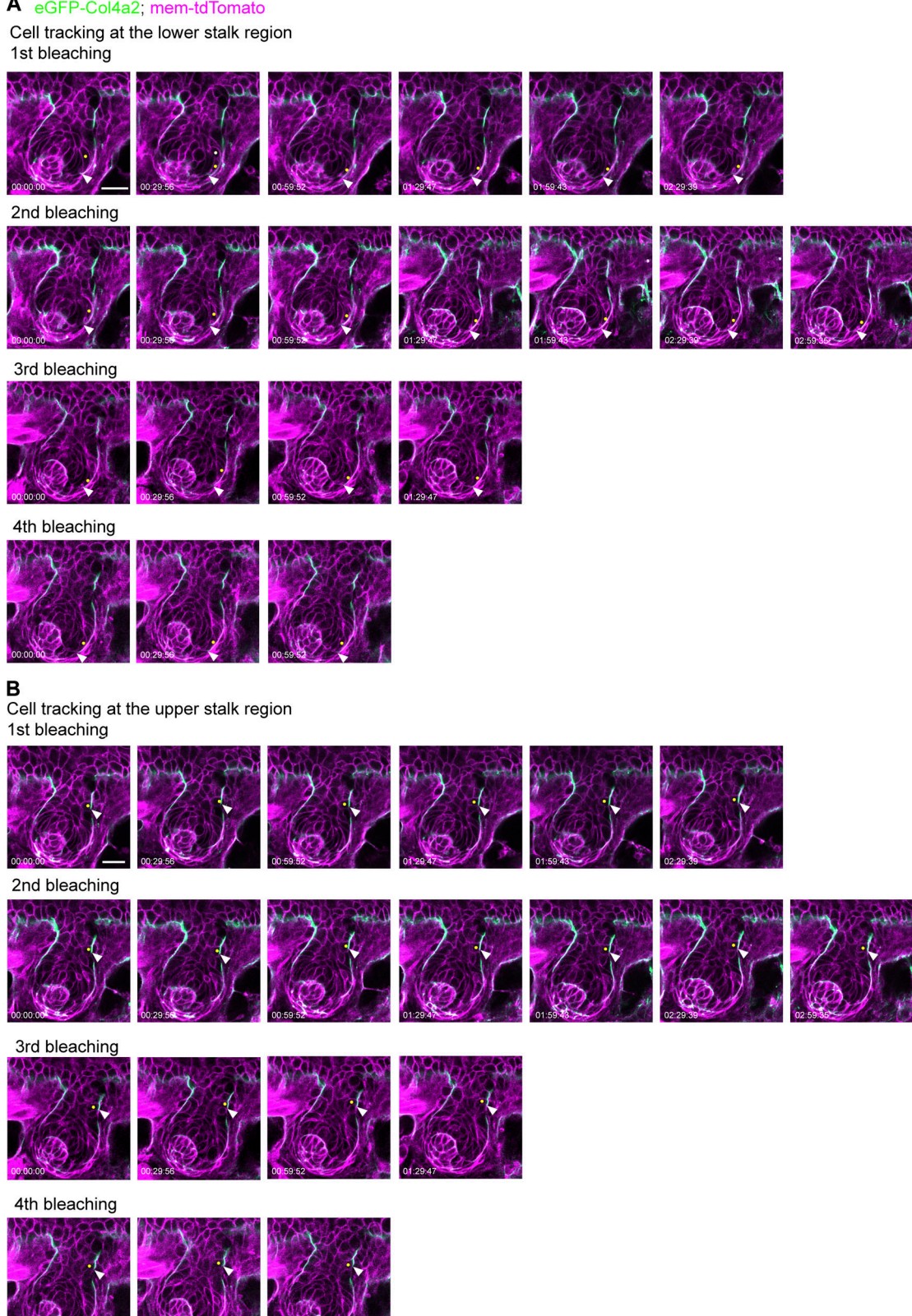

Figure S4. **Snapshot images of 3D time-lapse videos of developing hair follicles from *eGFP-Col4a2; mem-tdTomato* mice. (A and B)** Snapshot single z-plane images of cells and BMs tracking at 30-min intervals from 0 to 9 h 30 min. Cells and the BMs in the lower (A) and upper stalk regions (B) were tracked. The BM was re-photobleached several times to maintain a clear photobleached edge. The lower photobleached edge of the BM (indicated by white arrowheads) and epithelial basal progenitors initially located at this photobleached edge (indicated by yellow dots) were tracked. A white dot marks the other daughter cell following the division of the cell marked with a yellow dot. Scale bar: 20 μm.

Replicate No.2

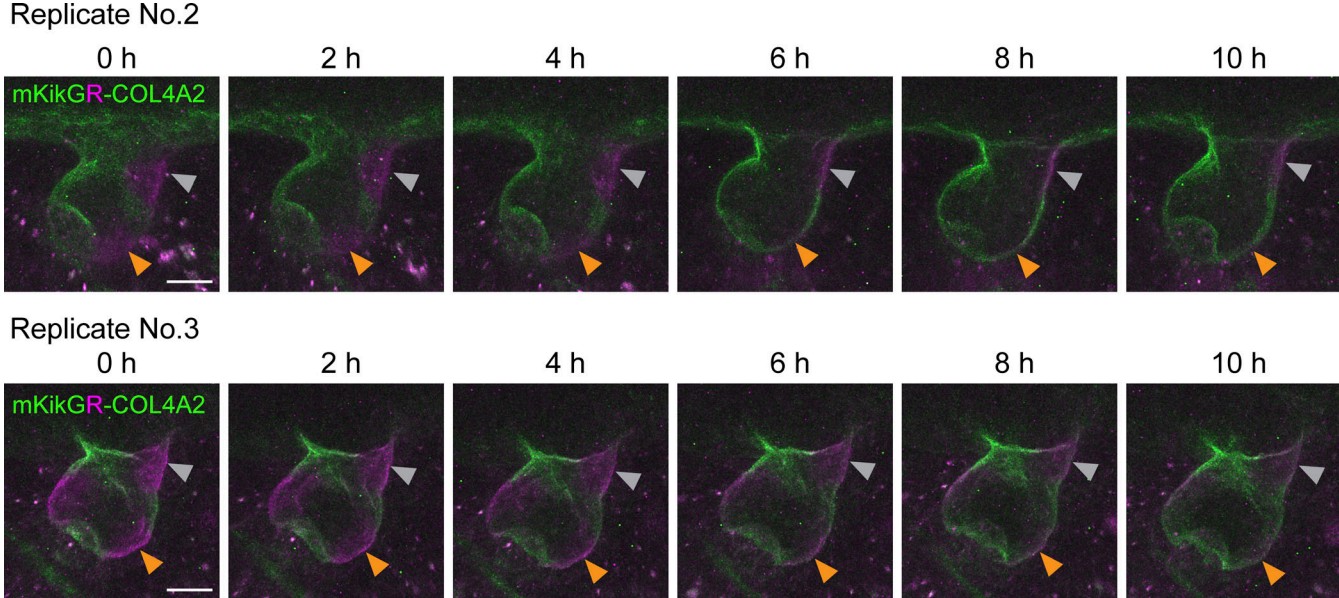

Figure S5.  **Imaging of mKikGR-COL4A2 in developing hair follicles.** Snapshot images of z-stack maximum projections of mKikGR-COL4A2 in developing hair follicles in embryonic skin explants from *mKikGR-Col4a2* mice at the indicated time points after photoconversion. Scale bar: 20 µm.

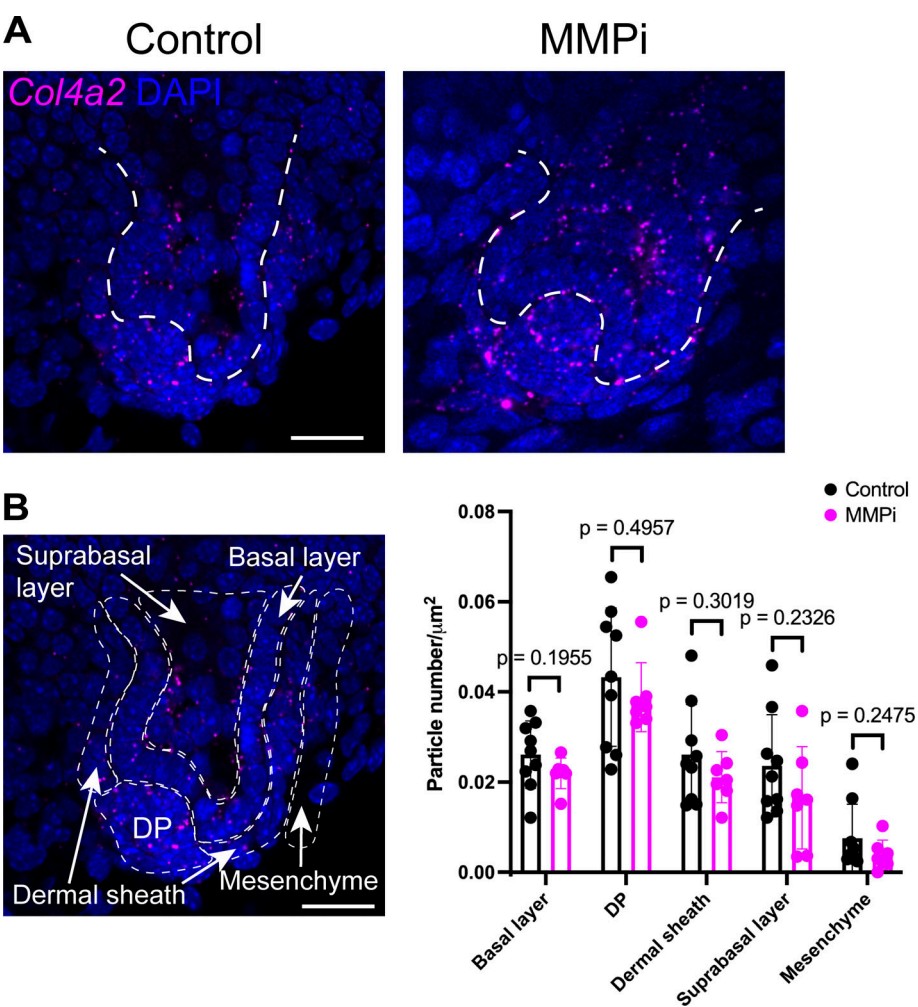

Figure S6. **Col4a2 mRNA expression in developing hair follicles. (A)** Confocal images of *Col4a2* mRNA expression in control and MMP inhibitor-treated developing hair follicles, obtained via RNAscope. The signals were detected 16 h after the addition of the MMP inhibitor batimastat. Scale bar: 20 μm. **(B)** Quantitative analysis of *Col4a2* mRNA expression in control and MMP inhibitor-treated developing hair follicles in different tissue regions. (*n* = 9 follicles from five explants, each from an independent experiment for the control, and *n* = 7 follicles from six explants, each from an independent experiment for the inhibitor-treated). Scale bar: 20 μm. Values: means ± SD. DP: dermal papilla. Two-tailed unpaired *t* tests were used. Data distribution was assumed to be normal, but this was not formally tested.

Video 1. **Representative time-lapse video capturing eGFP-COL4A2 signals in mouse embryonic dorsal skin explant cultures.** Representative time-lapse video of the 3D maximum projection of cultured skin explants from E12.5 embryos of *eGFP-Col4a2* mice, corresponding to Fig. 1 J. The BM was visualized with eGFP-COL4A2 (green). The BMs of hair follicles, interfollicular epidermis, and vessels are visualized. Live imaging was performed under a TCS SP8X (Leica) confocal microscope. Frames were collected approximately every 29 min and 24 s over the 19-h and 6-min time-lapse. The video is 13 s in total and displayed at ~3.08 frames per second.

Video 2. **Representative time-lapse video of FRAP experiments with eGFP-COL4A2 in developing hair follicles.** eGFP-COL4A2 signals (green) were photobleached in designated regions and signal recovery was monitored. Cultured skin explants from E12.5 embryos of *eGFP-Col4a2* mice were used, corresponding to Fig. 3 A. Live imaging was performed under a TCS SP8X (Leica) confocal microscope. Frames were collected approximately every 10 min and 7 s over the 5-h and 43-min time-lapse. The video is 37 s in total, with the first and last frames each held for 2 s to indicate regions of interest. The remaining frames are displayed at ~0.95 frames per second.

Video 3.   **Representative time-lapse video of pulse-chase experiments in the developing hair follicles of skin explants from *mKikGR-Col4a2* mouse embryos.** The green mKikGR-COL4A2 signals were photoconverted to red (magenta), and the changes in these mKikGR-COL4A2 signals were chased over ~19 h at 30 min intervals, corresponding to Fig. 3 D. Live imaging was performed under a TCS SP8X (Leica) confocal microscope. Frames were collected approximately every 28 min and 27 s during the 18-h and 57-min time-lapse. The video is 41 s in total and displayed at one frame per second.

Video 4.   **Representative time-lapse video of FRAP experiments with eGFP-COL4A2 in developing hair follicles under MMP inhibition.** EGFP-COL4A2 signals (green) were photobleached in designated regions and signal recovery was monitored under MMP inhibition with batimastat. Cultured skin explants from E12.5 embryos of *eGFP-Col4a2* mice were used, corresponding to Fig. 4 B, under MMP-inhibitor conditions. Live imaging was performed under a TCS SP8X (Leica) confocal microscope. Frames were collected approximately every 16 min and 13 s during the 18-h and 54-min time-lapse. The video is 1 min and 14 s in total, with the first and last frames each held for 2 s to indicate regions of interest. The remaining frames are displayed at one frame per second.

Video 5.   **Representative time-lapse video of daughter cells undergoing perpendicular or horizontal division during hair follicle development.** This time-lapse video shows tracked epithelial basal progenitors that undergo perpendicular or horizontal division in cultured skin explants from *eGFP-Col4a2; mem-tdTomato* mouse embryos. The BM and cell membrane were visualized with eGFP-COL4A2 (green) and mem-tdTomato (magenta), corresponding to Fig. 5 E. Live imaging was performed under an LSM980 (Carl Zeiss) with a two-photon laser unit Chameleon Discovery NX (Coherent). Frames were collected approximately every 30 min during the 2-h time lapse. The last frame of each part is displayed at 2 s per frame. The remaining frames are displayed at 0.67 frames per second.

**Provided online are Table S1 and Data S1. Table S1 lists key resources used in this study. Data S1 includes raw data used for quantification in this study.**

