## [Peer Review File · The Journal of Cell Biology]

An eGFP-Col4a2 mouse model reveals basement membrane dynamics underlying hair follicle morphogenesis

Duligengaowa Wuergenzhen, Eleonore Gindroz, Ritsuko Morita, Kei Hashimoto, Takaya Abe, Hiroshi Kiyonari, and Hironobu Fujiwara

Corresponding Author(s): Hironobu Fujiwara, RIKEN Center for Biosystems Dynamics Research

Review Timeline:

Submission Date:	2024-04-01
Editorial Decision:	2024-04-12
Revision Received:	2024-10-26
Editorial Decision:	2024-10-30
Revision Received:	2024-11-05

Monitoring Editor: Kenneth Yamada

Scientific Editor: Dan Simon

Transaction Report:

DOI: <https://doi.org/10.1083/jcb.202404003>

Revision 0

Review #1

1. Evidence, reproducibility and clarity:

Evidence, reproducibility and clarity (Required)

****Summary:****

Wuergezhen et al. generated two fluorescently tagged Col4a2 mouse models, including EGFP-Col4a2 and the photoconvertible mKikGR-Col4a2. They showed that tagged collagen IV get incorporated in the basement membranes of various tissues in the developing and adult mice, and fully support embryonic development, fertility, and physiological functions. The authors followed up investigating Col4a2 dynamics during embryonic development of the hair follicle. Using FRAP, they showed that the basement membrane expands with a faster rate near the tip region. This coincided with faster turnover rates of Col4a2 at the tip relative to the lower and upper stalk. In addition, the authors demonstrated that Col4a2 turnover depends on MMPs.

****Major comments:****

- Are the key conclusions convincing?

The major conclusions of the manuscript are convincing. These include that tagging collagen IV does not compromise its function, differential expansion of the basement membrane, differential turnover of the Col4a2, the MMP dependence for normal basement membrane expansion and turnover. However, some claims detailed below need to be clarified or addressed to improve the manuscript.

- Should the authors qualify some of their claims as preliminary or speculative, or remove them altogether?

In Figure 2B-C, the authors conclude that basement membrane expands at different rates, depending on the region with higher expansion rates near the hair follicle tip. From their methods, they conducted repetitive photobleaching cycles every 2-3 hours to maintain the bleached status of ROIs and tracked changes in basement membrane length. However, it sounds challenging to repetitively photobleach the same ROI in a dynamically expanding tissue, which may compromise the accuracy of length measurements. It would be helpful if the authors could provide movies corresponding to these experiments for clarity. In addition, this repetitive bleaching should be highlighted in the figure and figure legend, as readers could get confused about why there is no recovery here when comparing to the FRAP experiment.

In Lines 180-198 and Figure 2F-H, the authors interpreted cell movement as contributed by cell-autonomous vs basement membrane expansion, which is speculative. Another possibility is that cell movement was all autonomous, while basement membrane expansion was caused by mechanical stretching that was in turn caused by cell proliferation. This conclusion should be rephrased.

Following MMPi treatment, the authors found that basement membrane expansion was halted (Figure 4C-D). Yet, they later showed that hair follicles widened under MMPi treatment (Figure 4E-J). Under these conditions, I would have expected such widening to be accompanied by basement membrane expansion at the lower or upper stalk, which is not the case, at least for the first 7 hours (Figure 4C-D). How do the authors interpret this?

- Would additional experiments be essential to support the claims of the paper? Request additional experiments only where necessary for the paper as it is, and do not ask authors to open new lines of experimentation.

Considering the rapid expansion of BM near the tips, I wonder whether the authors have explored possible structural differences of the basement membrane in different regions? Is the basement membrane near the tips thinner and/or does it have microperforations as reported in other systems? Looking into this may further support their observations, not only in control conditions but also in MMPi treated follicles.

- Are the suggested experiments realistic in terms of time and resources? It would help if you could add an estimated cost and time investment for substantial experiments.

Yes, for possible structural changes, 3D rendering of fluorescence imaging as shown in Figure 1J may be sufficient.

Increasing sample number using existing mouse strains should also be feasible. It would take 2-3 weeks from setting up timed pregnant mice to imaging the embryonic skin explants. Adding image analysis and figure preparation, it should be doable within 1 month.

- Are the data and the methods presented in such a way that they can be reproduced?

In Figure 2D/ Figure 5C, it is unclear how ROIs corresponding to tip/lower/upper stalk are being drawn for quantification of Ki67+ cells.

The images in 2B and 4C, 3A and 4E, are identical. Images should not be re-used in figures. This raised the concern that probably not sufficient samples were imaged to have different representative images. It should also be clarified where the data was re-used, for example, the control data in 2C and 4D, 3B and 4B.

- Are the experiments adequately replicated and statistical analysis adequate?

A few experiments have low sample numbers while the data was quite variable. These include Figure 2G-H (n = 3 cells), Figure 3C (unknown sample number), Figure 4B (n = 3 hair follicles), Figure 5B (n = 3 hair follicles). More sample numbers (~10 total) should be included to solidify the findings.

For the mKikGR experiment in Figure 3C, quantification should be included. A ratiometric measurement of green/red fluorescence over time should be a good complementary way of demonstrating region-specific recovery.

In Figure 5B, the authors claim that the percentage of dividing cells in control follicles being

different from that of MMPi-treated follicles. How are they extracting these percentages? From the plots, control and MMPi-treated columns do not appear to be normalized as 100% to make such comparisons. Moreover, having two mean \pm SD in each column makes these data confusing to interpret. The authors should consider replotting their data either by combining their data into a unique population per conditions and reporting the percentages, or alternatively, they may consider splitting each of these columns into two (i.e., dividing vs. non-dividing cells), and comparing both conditions as ratios of dividing versus/non-dividing cells.

****Minor comments:****

- Specific experimental issues that are easily addressable.

Already addressed above.

- Are prior studies referenced appropriately?

I believe so.

- Are the text and figures clear and accurate?

All plots measuring basement membrane length are labeled as 'increase in BM length', even when in some cases the BM length is reduced. The authors should consider relabeling these as 'BM length change' or something similar.

The second and third paragraphs of the discussion are too long and should be condensed into a single paragraph.

- Do you have suggestions that would help the authors improve the presentation of their data and conclusions?

Use better color combination for visualization of multi-channel images. For instance, magenta/green is a better combination than red/green for the color blind. Color combinations are not consistent across figures in the manuscript.

Include movies for all data derived from live imaging.

Include statistical tests used for all plots, some are missing.

The authors should consider fitting their FRAP data in each condition and report percentages corresponding to the mobile and immobile fractions.

Examples of horizontal vs perpendicular cell division appear to be mislabeled in Video5.

2. Significance:

Significance (Required)

- Describe the nature and significance of the advance (e.g. conceptual, technical, clinical) for the field.

- The large molecular weight of extracellular proteins makes it challenging to generate genetically engineered versions of such proteins to investigate their function and dynamics in vivo. The present study has addressed such issues for Col4a2, a major component of the basement membrane. This study further provides insights towards the understanding of BM dynamics during embryonic organ development.

- Place the work in the context of the existing literature (provide references, where appropriate).

- Addressed above

- State what audience might be interested in and influenced by the reported findings.

- Cell and developmental biology, extracellular matrix biology, organ development and regeneration.

- Define your field of expertise with a few keywords to help the authors contextualize your point of view. Indicate if there are any parts of the paper that you do not have sufficient expertise to evaluate.

- Tissue morphogenesis, extracellular matrix

3. How much time do you estimate the authors will need to complete the suggested revisions:

Estimated time to Complete Revisions (Required)

(Decision Recommendation)

Less than 1 month

4. Review Commons values the work of reviewers and encourages them to get credit for their work. Select 'Yes' below to register your reviewing activity at Web of Science Reviewer Recognition Service (formerly Publons); note that the content of your review will not be visible on Web of Science.

Yes

Review #2

1. Evidence, reproducibility and clarity:

Evidence, reproducibility and clarity (Required)

The authors successfully generated a knock-in mouse line expressing fluorescence-tagged endogenous collagen IV. Homozygous mice exhibited normal survival with no apparent developmental abnormalities. Through imaging of the back skin of COL4A2-eGFP mice in ex vivo culture, the authors observed a faster expansion rate of the basement membrane (BM) at the growing tips of developing hair follicles, while the upper stalk region showed a slower BM expansion rate. Real-time imaging also revealed that the BM and basal epithelial cells moved in the same direction. Fluorescence recovery after photobleaching (FRAP) and the photoconvertible protein mKikGR experiments demonstrated a faster turnover rate of COL4A2 at the tips of the hair follicles. The authors used MMP inhibitors to suppress the recovery of COL4A2-eGFP, BM expansion, and observed abnormalities in the developmental process of hair follicle morphology. The findings are

novel and interesting, some aspects of how the experiments and quantifications were conducted should be clarified to allow the readers to fully appreciate the results.

****Major Comments:****

1. BM Expansion Statistics: Regarding the statistical analysis of BM expansion, is the length of the BM affected by the z-axis plane at the time of imaging? How is it ensured that the BM length measured at different time points corresponds to the same z-axis?
2. Cell Movement Statistics (Figure 2): In the statistical analysis of cell movement in Figure 2, the lack of other references raises questions about ensuring that the cells measured at different time points are the same cells.
3. The authors concluded that the inhibition of MMP delayed the recovery of COL4A2-eGFP after photobleaching, indicating the crucial role of MMP activity in the incorporation of COL4A2 into the BM. Does MMP inhibition lead to a reduction in the synthesis of the COL4A2 protein, thereby delaying the recovery of COL4A2 in the BM?

****Minor Comments:****

In Figure 1D, the representative image and the statistical graph are inconsistent. For example, in the upper stalk region, all basal epithelial cells have Ki67 signals according to the representative image.

2. Significance:

Significance (Required)

The authors successfully generated knock-in mice expressing fluorescence-tagged endogenous collagen IV. With this novel mouse model, researchers can directly observe the dynamic changes in the extracellular matrix of different organs in mammals during development and disease, revealing the potential roles of the extracellular matrix.

3. How much time do you estimate the authors will need to complete the suggested revisions:

Estimated time to Complete Revisions (Required)

(Decision Recommendation)

Less than 1 month

4. Review Commons values the work of reviewers and encourages them to get credit for their work. Select 'Yes' below to register your reviewing activity at Web of Science Reviewer Recognition Service (formerly Publons); note that the content of your review will not be visible on Web of Science.

Yes

Review #3

1. Evidence, reproducibility and clarity:

Evidence, reproducibility and clarity (Required)

Abe et al. generated multiple novel mouse model that follows and labels Col4a2 by fusing it with GFP. The creation of this mouse model allowed for the analysis of the basement membrane in real-time using live imaging of embryonic explant cultures. The combination of the mouse model with live imaging revealed new insights into how the basement membrane turnover occurs during early hair follicle development. After setting the stage the authors establish its utility with embryonic tissue, which revealed membrane buds that could be visualized with eGFP at different time intervals for 19h. The growth of the hair follicle bud during that time could be divided three distinct areas, the upper, lower stalks, and the tip. Where the tip and lower stalk increased in BM length, while the upper stalk decreased. BM length correlated with increased cellular replication. Using a double transgenic model system, they evaluated the differences between 'cell displacement', 'BM expansion', and 'cell autonomous movement'. Diving deeper with the Col4a2-mKiGR mouse model investigated Col4a2 turnover to reveal similar results mentioned above. Lastly MMP inhibitors were used to investigate how hair follicle growth is affected, which revealed inhibition of cellular replication and inhibition of Col4a2 turnover. These results inhibited hair follicle elongation which made the forming hair follicles thicker.

Overall, this is an interesting paper for hair follicle biologists since it investigates the early moments of the budding follicle in real time using a ex vivo culture model. In addition, there is additional appeal because the authors use live imaging the hair follicle neogenesis as a basement membrane model, which is interesting and novel. The manuscript is clearly written, and the data support the overall conclusions. My comments below are to help in clarity which may be used to develop clearer figures and additional text.

****Major:****

In Figure 2 for the cell displacement and expansion part. I found the figure and text confusing, particularly in regard to how the data shows the bleached edge of the BM moving alongside the cells. Could the authors make this clearer in the figure and in the writing of the text?

The authors created a Col4a2-mKikGR mouse line that is supposed to follow Col4a2 turnover. It is difficult to understand how the authors can claim that it was turnover with the explanation in the text of how the mouse model works. Could the authors write a better description of the mouse model in the text?

2. Significance:

Significance (Required)

The strengths of the paper are the use of a novel mouse model that can track collagen (Col4a2) with a tagged GFP and the live imaging model system. This has led to a novel and important knowledge on the interplay between the BM and cells. The limitation of the manuscript is that the entire manuscript utilizes a single protocol (live imaging) to investigate BM dynamics. This might be due to the sophisticated nature of the model systems, but it is a limitation.

To my knowledge the study is novel because of the mouse model to track Col4a2 dynamics with GFP. This model led to some interesting findings about hair follicle development in that different regions expand differently with the BM. This is good foundational knowledge using the latest state of the art techniques in live imaging of embryonic tissue culture systems.

The audience that would be interested in this manuscript is broad, which would span cell and developmental biology but also specialists in membrane and protein biology that can finally see in real time the dynamics of Col4a2 building skin and hair.

My expertise in in hair follicle development and repair and stem cell biology.

3. How much time do you estimate the authors will need to complete the suggested revisions:

Estimated time to Complete Revisions (Required)

(Decision Recommendation)

Between 1 and 3 months

No

Revision Plan

Manuscript number: RC-2023-02239

Corresponding author(s): Hironobu, Fujiwara

[The “revision plan” should delineate the revisions that authors intend to carry out in response to the points raised by the referees. It also provides the authors with the opportunity to explain their view of the paper and of the referee reports.]

The document is important for the editors of affiliate journals when they make a first decision on the transferred manuscript. It will also be useful to readers of the reprint and help them to obtain a balanced view of the paper.

*If you wish to submit a full revision, please use our "Full Revision" template. **It is important to use the appropriate template to clearly inform the editors of your intentions.**]*

1. General Statements [optional]

This section is optional. Insert here any general statements you wish to make about the goal of the study or about the reviews.

The goal of our study is to: (1) report the successful development of two fluorescently tagged *Col4a2* mouse models that fully support embryonic development and physiological functions, even when homozygous; (2) report real-time visualisation methods of COL4A2 and basement membrane structures in living developing organs; and (3) demonstrate the spatially different molecular and tissue dynamics of the basement membrane and its involvement in mammalian organogenesis. I appreciate that the goal of this study was acknowledged by the reviewers.

We would like to express our appreciation for the high-quality feedback received from our reviewers. We found the reviewers' comments to be both relevant and constructive, offering clear pathways for improvement that are feasible within a reasonable timeframe and cost constraints, with two exceptions explained in the following sections. Moreover, we are committed to addressing these comments thoroughly in the revision process.

In light of recent publications, we wish to respectfully acknowledge the contributions of a similar study establishing a mouse model, *mTurq2-Col4a1* (Jones et al., *Journal of Cell Biology* 2024, PMID: 37808687, doi: <https://doi.org/10.1101/2023.09.27.559396>). The work by Jones et al. offers valuable insights into the dynamics of mammalian basement membranes and represents a significant step forward in the field. However, it has been noted that homozygous viability is totally compromised, as evidenced by the absence of homozygous pups from numerous heterozygous crosses, suggesting that the fusion COL4A1 protein cannot substitute endogenous COL4A1 functions. In contrast, our study introduced genes encoding a fluorescent protein in the *Col4a2* gene (and the *Col4a1* gene, though not discussed in our current study) in

different regions and employing a different strategy, and demonstrated normal growth in homozygous mice. This distinction not only demonstrates the functional viability of the mouse model, but also greatly broadens the scope for future research into embryonic development, physiological functions, and in vitro models such as organoids. Furthermore, our findings on dynamic basement membrane remodelling and its involvement in hair follicle organogenesis, which were not addressed in the Jones et al. study, add new dimensions to the understanding of this process. In our revised manuscript, we intend to reference Jones et al. with the utmost respect, aiming to contextualise our findings within the broader landscape of basement membrane research.

We believe these distinctions in the mouse models and the scope of functional investigations offer substantial contributions to the field. We hope this clarification highlights the unique value and potential impact of our study.

2. Description of the planned revisions

Insert here a point-by-point reply that explains what revisions, additional experimentations and analyses are planned to address the points raised by the referees.

Reviewers' comments:

Reviewer #1

Summary:

Wuergezhen et al. generated two fluorescently tagged Col4a2 mouse models, including EGFP-Col4a2 and the photoconvertible mKikGR-Col4a2. They showed that tagged collagen IV get incorporated in the basement membranes of various tissues in the developing and adult mice, and fully support embryonic development, fertility, and physiological functions. The authors followed up investigating Col4a2 dynamics during embryonic development of the hair follicle. Using FRAP, they showed that the basement membrane expands with a faster rate near the tip region. This coincided with faster turnover rates of Col4a2 at the tip relative to the lower and upper stalk. In addition, the authors demonstrated that Col4a2 turnover depends on MMPs.

Response:

The reviewer referred to our mouse models as “EGFP-Col4a2” and “mKikGR-Col4a2”, thereby placing the names of the fluorescent proteins before Col4a2, suggesting that these designations follow the appropriate naming conventions. Thus, we consulted the RIKEN Bioresource Research Center about the proper naming of our mouse lines. In accordance with their guidance, we have chosen to name them “eGFP-Col4a2” and “mKikGR-Col4a2”, respectively. We appreciate this reviewer’s attention in this area.

Revision Plan

Major comments:

- Are the key conclusions convincing?

- The major conclusions of the manuscript are convincing. These include that tagging collagen IV does not compromise its function, differential expansion of the basement membrane, differential turnover of the Col4a2, the MMP dependence for normal basement membrane expansion and turnover. However, some claims detailed below need to be clarified or addressed to improve the manuscript.

- Should the authors qualify some of their claims as preliminary or speculative, or remove them altogether?

- In Figure 2B-C, the authors conclude that basement membrane expands at different rates, depending on the region with higher expansion rates near the hair follicle tip. From their methods, they conducted repetitive photobleaching cycles every 2-3 hours to maintain the bleached status of ROIs and tracked changes in basement membrane length. However, it sounds challenging to repetitively photobleach the same ROI in a dynamically expanding tissue, which may compromise the accuracy of length measurements. It would be helpful if the authors could provide movies corresponding to these experiments for clarity. In addition, this repetitive bleaching should be highlighted in the figure and figure legend, as readers could get confused about why there is no recovery here when comparing to the FRAP experiment.

Response:

We will prepare supplementary videos or figures to show the process of repetitive photobleaching (see preliminary figure, Figure RP1, below).

These videos or figures will demonstrate that we have re-photobleached at a timepoint in which the border of photobleached and non-photobleached basement membrane regions are distinguishable from the difference in their fluorescent intensities. We adjusted the ROI size and position for each photobleaching session according to the position of the border of photobleached and non-photobleached basement membrane regions. A detailed explanation regarding the methods for re-photobleaching and measurement of basement membrane expansion will be included in Methods (please refer to our response to the first major comment of Reviewer #2). We will also highlight these procedures in the figure and the figure legend.

Revision Plan

Revision Plan

Figure RP1. Representative re-photobleaching procedures for the measurement of basement membrane expansion.

A. A z-plane exhibiting sharp and narrow signals from the basement membrane within the targeted area was selected for photobleaching. A ROI (indicated by a rectangular box) was set on the targeted z-plane for each region. The same photobleaching procedure was applied to each ROI. At intervals of 2 hours and 30 minutes culture following each photobleaching session, the edges of the photobleached areas were identified (distinguishable edges were indicated by arrowheads) and a new ROI was drawn along the identified photobleached edges. Additional z-planes are displayed only if some target areas in the main z-plane do not show sharp eGFP-COL4A2 signals.

B. These images show the procedures used to quantify basement membrane length changes. Measurements were conducted in different z-planes to determine the length of basement membrane in the lower stalk region (upper panels) and in the tip region (lower panels). Scale bar: 20 μm .

- In Lines 180-198 and Figure 2F-H, the authors interpreted cell movement as contributed by cell-autonomous vs basement membrane expansion, which is speculative. Another possibility is that cell movement was all autonomous, while basement membrane expansion was caused by mechanical stretching that was in turn caused by cell proliferation. This conclusion should be rephrased.

Response:

We appreciate the reviewer's insightful suggestion that cell movement might be entirely autonomous, with the expansion of the basement membrane resulting from mechanical stretching due to cell proliferation. This perspective will be discussed in our revised manuscript. Moreover, we recognise the difficulty in defining the range of "cell autonomous" systems within the context of our research. The term "cell autonomous" generally refers to effects or behaviours that originate within a cell, independent of external influences or interactions with other cells. However, complex interplay between cells and their microenvironment, particularly in a process such as basement membrane expansion, challenges the simplicity of this definition. Thus, we will undertake a careful re-examination of the term "cell autonomous" and rephrase our observations, including whether or not we use the term "cell autonomous".

- Following MMPi treatment, the authors found that basement membrane expansion was halted (Figure 4C-D). Yet, they later showed that hair follicles widened under MMPi treatment (Figure 4E-J). Under these conditions, I would have expected such widening to be accompanied by basement membrane expansion at the lower or upper stalk, which is not the case, at least for the first 7 hours (Figure 4C-D). How do the authors interpret this?

Response:

We appreciate the reviewer's valuable comment. To clarify the results of the MMPi experiments and to prevent potential misinterpretation, we will improve our manuscript with additional explanations. First, while the procedure for the MMPi experiments is described in detail in the

Methods section, we recognise the need for greater clarity. Therefore, we will include both a comprehensive description (see below) and a schematic illustration of the MMPi experimental procedure (Figure RP2A) in the Results section and in the appropriate figures.

Following the addition of MMPi, skin explants were cultured for 18 hours prior to a 20-hour live imaging session. This preparation ensured the MMPi's effects were uniformly distributed throughout the tissue. During this imaging period, we observed that MMPi treatment effectively reduced the incorporation of eGFP-COL4A2 and the expansion of the basement membrane (Figure 4A–D). Additionally, in MMPi treated hair follicles, elongation was halted, while significant elongation was observed in control hair follicles (Figure RP2B–E). Although the expansion of the basement membrane in the lower and upper stalk regions was stopped, the basement membrane of the tip region still expand a little (Figure 4D). Therefore, tissue deformation through remodeling and expansion of the basement membrane cannot be ruled out during the imaging period (0 h to 20 h).

Furthermore, it is important to note that the width of the hair follicle had already increased prior to the imaging period (Figure RP2D). These results indicate that MMPi treatment initially enables the widening of the hair follicle. An early stage of MMP inhibition or weak inhibition might permit the basement membrane to expand towards the direction of the perimeter, thereby leading to the formation of widened hair follicles. These more detailed data analysis, explanation and interpretation will be included in the revised version of the manuscript to enhance the understanding of the data.

Figure RP2. Analysis of hair follicle morphology in control and MMPi-treated samples over time.

- A. A schematic illustration of the experimental procedure of the MMP inhibition.
 B. Measurement of hair follicle (HF) length in control and MMPi-treated groups at the 0-hour time point.

Revision Plan

- C. Comparison of the relative change in hair follicle length between the 0-hour and 20-hour time points in both control and MMPi-treated conditions.
 - D. Measurement of hair follicle width in control and MMPi-treated groups at the 0-hour time point.
 - E. Comparison of the relative change in hair follicle width between the 0-hour and 20-hour time points in control and MMPi-treated conditions.
- (n = 8 hair follicles from 7 independent explants/experiments for the control, and n = 6 hair follicles from 3 independent explants/experiments for the inhibitor-treated condition). Values are presented as mean \pm SD. A two-tailed unpaired t-test was used.

- Would additional experiments be essential to support the claims of the paper? Request additional experiments only where necessary for the paper as it is, and do not ask authors to open new lines of experimentation.

- Considering the rapid expansion of BM near the tips, I wonder whether the authors have explored possible structural differences of the basement membrane in different regions? Is the basement membrane near the tips thinner and/or does it have microperforations as reported in other systems? Looking into this may further support their observations, not only in control conditions but also in MMPi treated follicles.

Response:

We will employ fluorescent microscopy to measure the thickness of the basement membrane in different tissue regions. We will also perform wholmount immunofluorescent imaging to investigate the presence of microperforations.

- Are the suggested experiments realistic in terms of time and resources? It would help if you could add an estimated cost and time investment for substantial experiments.

- Yes, for possible structural changes, 3D rendering of fluorescence imaging as shown in Figure 1J may be sufficient.

- Increasing sample number using existing mouse strains should also be feasible. It would take 2-3 weeks from setting up timed pregnant mice to imaging the embryonic skin explants. Adding image analysis and figure preparation, it should be doable within 1 month.

Response:

We would like to provide an estimated cost and time investment for suggested experiments. Typically, a high time-resolution continuous live imaging yields data from only one hair follicle per culture of embryonic skin tissue. This process requires the use of a two-photon microscope for up to 24 hours. Thus, to obtain images of five hair follicles (yielding five biological replicates), for example, at least five timed mating events on five different days are required, in alignment with the availability of the two-photon microscope. Occasionally, live imaging sessions must be terminated prematurely if the developing hair follicles move out of the imaging fields, particularly

Revision Plan

along the z-axis. Consequently, increasing the number of replicates for live imaging data demands a significant number of mouse mating and imaging sessions. Considering the cost and time investment for live imaging, we would like to propose ending the acquisition of replicate data once we obtain high quality data with low variability, even if this occurs before reaching the number of replicates recommended by the reviewer. In addition, some of the current data with a high degree of variation resulted from a slight difference in developmental stages of the hair follicle used, rather than a variation in the trend of differences in measuring results between hair follicle regions. Therefore, we will mention this potential cause of variation in the revised manuscript and/or normalise the data.

- Are the data and the methods presented in such a way that they can be reproduced?

- In Figure 2D/ Figure 5C, it is unclear how ROIs corresponding to tip/lower/upper stalk are being drawn for quantification of Ki67+ cells.

Response:

We will include ROI indicators to clarify the regions measured.

- The images in 2B and 4C, 3A and 4E, are identical. Images should not be re-used in figures. This raised the concern that probably not sufficient samples were imaged to have different representative images. It should also be clarified where the data was re-used, for example, the control data in 2C and 4D, 3B and 4B.

Response:

We appreciate the reviewer's suggestion. We will address this issue by replacing the identical images as suggested. We would also like to note that the original manuscript does acknowledge the reuse of certain data. For instance, the legend for Figure 4C states that, "Control data from this experiment are shown in Figure 2B," to inform readers of intentional data reuse. This approach was taken to maintain consistency across related experiments, although we do agree with the reviewer's concern.

- Are the experiments adequately replicated and statistical analysis adequate?

- A few experiments have low sample numbers while the data was quite variable. These include Figure 2G-H (n = 3 cells), Figure 3C (unknown sample number), Figure 4B (n = 3 hair follicles), Figure 5B (n = 3 hair follicles). More sample numbers (~10 total) should be included to solidify the findings.

Response:

We will conduct additional experiments to increase the number of replicates. Based on the reasons described above, we propose concluding the acquisition of replicate data once we have obtained high quality data with low variability, even if this occurs before reaching the number of replicates recommended by the reviewer.

Revision Plan

- For the *mKikGR* experiment in Figure 3C, quantification should be included. A ratiometric measurement of green/red fluorescence over time should be a good complementary way of demonstrating region-specific recovery.

Response:

We attempted the ratiometric measurement of green/red fluorescence for the experiment in Figure 3C using single z-plane images, but encountered significant variability due to low fluorescent intensity, low signal-to-noise ratio, and high susceptibility to photobleaching of *mKikGR*. Therefore, instead of showing unreliable quantification data, we will include replicate maximum projection image data of entire hair follicles, in supplementary figures as shown below (Figure RP 3).

Replicate No.2

Replicate No.3

Figure RP3. Maximum projection images of *mKikGR-Col4a2* in developing hair follicles of embryonic skin explants.

Lower stalk (orange arrowheads) and follicular–interfollicular junction (grey arrowheads) regions were photoconverted, and signal intensity changes were monitored for 10 hours.

Scale bar: 20 μm .

- In Figure 5B, the authors claim that the percentage of dividing cells in control follicles being different from that of MMPi-treated follicles. How are they extracting these percentages? From the plots, control and MMPi-treated columns do not appear to be normalized as 100% to make such comparisons. Moreover, having two mean \pm SD in each column makes these data confusing to interpret. The authors should consider replotting their data either by combining their data into a unique population per conditions and reporting the percentages, or alternatively, they may consider splitting each of these columns into two (i.e., dividing vs. non-dividing cells), and comparing both conditions as ratios of dividing versus/non-dividing cells.

Revision Plan

Response:

We have replotted the data to show dividing and non-dividing cells under each condition (see the updated graphs below). In addition, we will increase the replicates for this experiment.

Figure RP4. Quantification of the cell division of epithelial basal progenitors in the experiments shown in Figure 5A.

F. Percentage of dividing cells in the control and under MMPi conditions.

G. Percentage of non-dividing cells in the control and under MMPi conditions.

Minor comments:

- Specific experimental issues that are easily addressable.
- *Already addressed above.*

- Are prior studies referenced appropriately?
- *I believe so.*

- Are the text and figures clear and accurate?
- *All plots measuring basement membrane length are labeled as 'increase in BM length', even when in some cases the BM length is reduced. The authors should consider relabeling these as 'BM length change' or something similar.*

Response:

We will relabel them appropriately, as suggested.

- *The second and third paragraphs of the discussion are too long and should be condensed into a single paragraph.*

Response:

We will condense these two paragraphs into a single paragraph, as suggested.

- Do you have suggestions that would help the authors improve the presentation of their data and conclusions?
- *Use better color combination for visualization of multi-channel images. For instance, magenta/green is a better combination than red/green for the color blind. Color combinations are not consistent across figures in the manuscript.*

Revision Plan

Response:

We will change the colour combinations to magenta/green for all figures, as suggested.

- Include movies for all data derived from live imaging.

Response:

We will submit all data derived from live imaging to the SSBD public bioimage database (<https://ssbd.riken.jp/database/>).

- Include statistical tests used for all plots, some are missing.

Response:

We will include the statistical tests employed for all plots.

- The authors should consider fitting their FRAP data in each condition and report percentages corresponding to the mobile and immobile fractions.

Response:

In response to the reviewer's request, we have extensively evaluated the feasibility of fitting our FRAP data to calculate the mobile and immobile fractions of eGFP-COL4A2 in the basement membrane. Our observations indicate a very slow signal recovery and the lack of a clear plateau within the experimental timeframe. This slow recovery, coupled with dynamic changes in the tissue structure and the potential increase and decrease in the total amount of eGFP-COL4A2 in the basement membrane, underscores the non-static nature of our system. These factors introduce significant variability in molecular dynamics, thereby rendering the application of standard fitting models, which presuppose stable conditions and experimental system, inappropriate for our data.

Moreover, the unique protein dynamics of COL4A2, including its very limited diffusibility and slow turnover rates within the basement membrane, necessitate prolonged live-imaging sessions, potentially ranging from hours to days, to capture the full recovery process. Such prolonged observation periods are required to accurately capture the plateau of eGFP-COL4A2 recovery intensity. However, tracking the photobleached basement membrane zone throughout the recovery period becomes increasingly challenging due to morphological changes in embryonic skin tissues, including that of the basement membrane. The original photobleached zone is located by identifying the boundary between the bleached and non-photobleached basement membrane areas. However, once the fluorescent recovery rate exceeds approximately 50%, distinguishing this boundary becomes problematic, leading to difficulties in accurately measuring the fluorescence recovery rates.

Given these challenges, particularly the difficulty in maintaining continuous measurements to achieve a recovery plateau, we have refrained from calculating the mobile and immobile fractions in our FRAP analysis. To ensure consistency across target ROIs, we applied the same laser irradiation programme for photobleaching within each ROI from different

Revision Plan

regions. Our experiments underscore the need for developing novel analytical frameworks to better accommodate the dynamic and complex nature of biological systems such as ours.

To delineate the significant difference in signal intensity recovery between conditions, we plan to extract data from specific timepoints and present this subset separately to highlight statistically significant differences.

- *Examples of horizontal vs perpendicular cell division appear to be mislabeled in Video5.*

Response:

We will correct this error, accordingly.

Reviewer #1 (Significance (Required)):

- Describe the nature and significance of the advance (e.g. conceptual, technical, clinical) for the field.

- *The large molecular weight of extracellular proteins makes it challenging to generate genetically engineered versions of such proteins to investigate their function and dynamics in vivo. The present study has addressed such issues for Col4a2, a major component of the basement membrane. This study further provides insights towards the understanding of BM dynamics during embryonic organ development.*

- Place the work in the context of the existing literature (provide references, where appropriate).

- *Addressed above*

- State what audience might be interested in and influenced by the reported findings.

- *Cell and developmental biology, extracellular matrix biology, organ development and regeneration.*

- Define your field of expertise with a few keywords to help the authors contextualize your point of view. Indicate if there are any parts of the paper that you do not have sufficient expertise to evaluate.

- *Tissue morphogenesis, extracellular matrix*

Reviewer #2

The authors successfully generated a knock-in mouse line expressing fluorescence-tagged endogenous collagen IV. Homozygous mice exhibited normal survival with no apparent developmental abnormalities. Through imaging of the back skin of COL4A2-eGFP mice in ex vivo culture, the authors observed a faster expansion rate of the basement membrane (BM) at the growing tips of developing hair follicles, while the upper stalk region showed a slower BM expansion rate. Real-time imaging also revealed that the BM and basal epithelial cells moved in the same direction. Fluorescence recovery after photobleaching (FRAP) and the

Revision Plan

photoconvertible protein mKikGR experiments demonstrated a faster turnover rate of COL4A2 at the tips of the hair follicles. The authors used MMP inhibitors to suppress the recovery of COL4A2-eGFP, BM expansion, and observed abnormalities in the developmental process of hair follicle morphology. The findings are novel and interesting, some aspects of how the experiments and quantifications were conducted should be clarified to allow the readers to fully appreciate the results.

Major Comments:

1. BM Expansion Statistics: Regarding the statistical analysis of BM expansion, is the length of the BM affected by the z-axis plane at the time of imaging? How is it ensured that the BM length measured at different time points corresponds to the same z-axis?

Response:

To measure the length of the basement membrane in the same tissue regions over time, the first step is to identify a z-plane characterised by sharp and narrow signals from the basement membrane in the targeted area. These distinct signals suggest that this z-plane successfully captures the basement membrane oriented at or near a vertical angle (near the mid-line of the hair follicle). Conversely, a z-plane that is misaligned (aligned with tilted basement membranes) can be identified by imaging the presence of the basement membrane positioned obliquely to the scanning plane, resulting in a reduction in the clarity and sharpness of the signals from the basement membrane. These characteristics of the basement membrane's appearance make it possible to locate the z-position near the originally measured basement membrane region within our experimental period. For example, as shown in the figure below (Figure RP 5), the basement membrane of the lower stalk (orange arrowheads) and upper stalk (grey arrowheads) can be clearly observed as a thin, sharp, line-like structure at 0 h of the Single z-plane image, whereas they exhibit blurred and thicker structures in a different z-plane named the Single z'-plane. On the other hand, the basement membrane of the tip region (blue arrowheads) shows a blurred and thicker structure in the Single z-plane image, whereas it exhibits a clear thin basement membrane in the Single z'-plane. After 7 hours, the clear basement membrane of each compartment can be observed in the z-plane, indicated as "Single z-plane", which can be considered as the z-plane located near the original basement membrane region.

Furthermore, 3D-view images in Figure RP5 maintains a clear photobleached edge during the 7-hour experimental period, even if it contains several different z-plane images. This is a result of having photobleached 10 ± 3 continuous z-planes (with $1 \mu\text{m}$ interval) with the same ROI, leading to the formation of a 3D cuboidal bleached region. Therefore, even if the target z-position is slightly displaced during the measurement analysis, the results of the measurement of the basement membrane length remain constant because the dynamics of the basement membrane do not appear to drastically change within the circumferential location of the hair follicle. These new data and explanations will be included in the revised manuscript.

Figure RP5. Selection and monitoring of initial z-plane for photobleaching to analyse basement membrane expansion.

A 3D image of a developing hair follicle was captured, from which a single z-plane displaying sharp and narrow signals from the basement membrane within the targeted area was selected. Note that the basement membranes in the tip (blue arrowheads), lower stalk (orange arrowheads), and follicular–interfollicular junction (grey arrowheads) regions are indicated. The procedure included photobleaching both edges of the identified basement membrane zones. After a 7-hour culture period that included the use of repeated photobleaching, targeted basement membrane zones displaying sharp and narrow signals were identified for measuring the expansion of the basement membrane. Scale bar: 20 μ m.

2. *Cell Movement Statistics (Figure 2): In the statistical analysis of cell movement in Figure 2, the lack of other references raises questions about ensuring that the cells measured at different time points are the same cells.*

Response:

Revision Plan

We will provide corresponding time-lapse image figures and/or videos to demonstrate that the same cells can be accurately tracked over time.

3. The authors concluded that the inhibition of MMP delayed the recovery of COL4A2-eGFP after photobleaching, indicating the crucial role of MMP activity in the incorporation of COL4A2 into the BM. Does MMP inhibition lead to a reduction in the synthesis of the COL4A2 protein, thereby delaying the recovery of COL4A2 in the BM?

Response:

Due to the slow incorporation and turnover of eGFP-COL4A2 under MMPi conditions, a possible feedback reaction that reduces the synthesis of the COL4A2 protein by sensing the retarded COL4A2 dynamics or excess amount of COL4A2 may occur. As it is difficult to measure the amount of cellular synthesis of COL4A2 protein in vivo, we will examine the changes in the mRNA expression levels of *Col4a2* under MMPi conditions as a proxy for the synthesis of the COL4A2 protein. We plan to conduct RNAscope experiments on tissue sections or wholemounts.

Minor Comments:

In Figure 1D, the representative image and the statistical graph are inconsistent. For example, in the upper stalk region, all basal epithelial cells have Ki67 signals according to the representative image.

Response:

Thank you for bringing this to our attention. We acknowledge that the percentage of Ki67-positive cells depicted in the statistical graph of Figure 2E show significant variability, and there appears to be a lack of consistency between the representative image in Figure 2D and the statistical graph. This inconsistency may be attributed to the variable signal intensities observed in our Ki67 staining, including instances of very faint staining, which could lead to variability in the detection of Ki67-positive cells across samples. Furthermore, since Ki67 is expressed throughout nearly the entire mitotic process, it does not precisely indicate proliferating cells within a specific time frame. Consequently, for the purpose of directly comparing regions with high mitotic activity and specifically measuring cell division rates, we will employ phosphor-histone H3 (pH3) staining and/or EdU incorporation assays. These methods will enable us to accurately detect mitotic cells by identifying cells in the M-phase or transitioning from the S-phase to mitosis.

Reviewer #2 (Significance (Required)):

The authors successfully generated knock-in mice expressing fluorescence-tagged endogenous collagen IV. With this novel mouse model, researchers can directly observe the dynamic changes in the extracellular matrix of different organs in mammals during development and disease, revealing the potential roles of the extracellular matrix.

Revision Plan

Reviewer #3

Reviewer #3 (Evidence, reproducibility and clarity (Required)):

Abe et al. generated multiple novel mouse model that follows and labels Col4a2 by fusing it with GFP. The creation of this mouse model allowed for the analysis of the basement membrane in real-time using live imaging of embryonic explant cultures. The combination of the mouse model with live imaging revealed new insights into how the basement membrane turnover occurs during early hair follicle development. After setting the stage the authors establish its utility with embryonic tissue, which revealed membrane buds that could be visualized with eGFP at different time intervals for 19h. The growth of the hair follicle bud during that time could be divided three distinct areas, the upper, lower stalks, and the tip. Where the tip and lower stalk increased in BM length, while the upper stalk decreased. BM length correlated with increased cellular replication.

Using a double transgenic model system, they evaluated the differences between 'cell displacement', 'BM expansion', and 'cell autonomous movement'. Diving deeper with the Col4a2-mKiGR mouse model investigated Col4a2 turnover to reveal similar results mentioned above. Lastly MMP inhibitors were used to investigate how hair follicle growth is affected, which revealed inhibition of cellular replication and inhibition of Col4a2 turnover. These results inhibited hair follicle elongation which made the forming hair follicles thicker.

Overall, this is an interesting paper for hair follicle biologists since it investigates the early moments of the budding follicle in real time using a ex vivo culture model. In addition, there is additional appeal because the authors use live imaging the hair follicle neogenesis as a basement membrane model, which is interesting and novel. The manuscript is clearly written, and the data support the overall conclusions. My comments below are to help in clarity which may be used to develop clearer figures and additional text.

Major:

In Figure 2 for the cell displacement and expansion part. I found the figure and text confusing, particularly in regard to how the data shows the bleached edge of the BM moving alongside the cells. Could the authors make this clearer in the figure and in the writing of the text?

Response:

We will provide time-lapse image figures and/or videos to show this process in a clearer fashion, as well as indicate the tip of the photobleached edge of the BM moving alongside the cells.

The authors created a Col4a2-mKikGR mouse line that is supposed to follow Col4a2 turnover. It is difficult to understand how the authors can claim that it was turnover with the explanation in the text of how the mouse model works. Could the authors write a better description of the mouse model in the text?

Revision Plan

Response:

We will provide a more detailed explanation of the mKikGR mouse model and the experimental procedures that enabled us to assess the turnover of COL4A2. This model facilitates our understanding of whether: (1) the pre-existing COL4A2 protein (photoconverted to red mKikGR-COL4A2) either disperses/disappear from the specified basement membrane region (ROI) or remains in the same place; (2) new COL4A2 proteins (indicated by the influx of green mKikGR-COL4A2) are incorporated into the ROI; and (3) the movement of these proteins is driven by the processes of synthesis, assembly, and degradation, rather than by the diffusion of COL4A2 from adjacent basement membrane areas.

For example, at the tip of the hair follicle, if the photo-converted red mKikGR-COL4A2 signals persist alongside an increase in the green mKikGR-COL4A2 signals, it suggests minimal turnover, with incorporation being the predominant activity. However, we observed a significant decrease in the red signal as the green signal increased. We also observed that the transition from the red to the green signal within the ROI was relatively uniform, suggesting that the changes were not due to diffusion from the photobleached edges of the ROI. These results suggest active turnover of COL4A2 protein in the basement membrane, as opposed to mere incorporation or diffusion from the surrounding areas. Thus, this approach enabled us to discern the dynamics of COL4A2 turnover in the basement membrane.

Reviewer #3 (Significance (Required)):

The strengths of the paper are the use of a novel mouse model that can track collagen (Col4a2) with a tagged GFP and the live imaging model system. This has led to a novel and important knowledge on the interplay between the BM and cells. The limitation of the manuscript is that the entire manuscript utilizes a single protocol (live imaging) to investigate BM dynamics. This might be due to the sophisticated nature of the model systems, but it is a limitation.

To my knowledge the study is novel because of the mouse model to track Col4a2 dynamics with GFP. This model led to some interesting findings about hair follicle development in that different regions expand differently with the BM. This is good foundational knowledge using the latest state of the art techniques in live imaging of embryonic tissue culture systems.

The audience that would be interested in this manuscript is broad, which would span cell and developmental biology but also specialists in membrane and protein biology that can finally see in real time the dynamics of Col4a2 building skin and hair.

My expertise in in hair follicle development and repair and stem cell biology.

3. Description of the revisions that have already been incorporated in the transferred manuscript

Please insert a point-by-point reply describing the revisions that were already carried out and included in the transferred manuscript. If no revisions have been carried out yet, please leave this section empty.

4. Description of analyses that authors prefer not to carry out

Please include a point-by-point response explaining why some of the requested data or additional analyses might not be necessary or cannot be provided within the scope of a revision. This can be due to time or resource limitations or in case of disagreement about the necessity of such additional data given the scope of the study. Please leave empty if not applicable.

- 1) Based on the reasons described in our point-by-point response to Reviewer #1, we have refrained from calculating the mobile and immobile fractions in our FRAP analysis.
- 2) Considering the cost and time investment of live imaging, we would like to propose ending the acquisition of replicate data once we have obtained high quality data with low variability, even if this occurs before reaching the number of replicates recommended by Reviewer #1 (please see our point-by-point response for additional information).

April 12, 2024

Re: JCB manuscript #202404003T

Dr. Hironobu Fujiwara
RIKEN Center for Biosystems Dynamics Research
2-2-3 Minatojima Minamimachi
Chuo-ku
Kobe 6500047
Japan

Dear Dr. Fujiwara,

Thank you for submitting your manuscript entitled "Col4a2-eGFP mouse model reveals the molecular and functional dynamics of basement membrane remodeling in hair follicle morphogenesis." We have assessed the manuscript as well as the reviews from Review Commons and we invite you to submit a revised manuscript as outlined in your revision plan.

We agree that your FRAP analysis cannot allow definitive calculation of mobile and immobile fractions. Nevertheless, because of the confusion in the developmental biology literature about the rate of turnover of collagen IV in *Drosophila*, etc. versus in mouse, the data in Fig. 4B appears to deserve further discussion. For example, would you consider indicating some numbers, e.g. possibly turnover of half by 2.5 hours at the tip versus significantly higher stability elsewhere (lower stalk), e.g., retention of 75% at 4 hours? We also agree with your plans for limiting the numbers of replicates for these complex mouse experiments to practical limits while still providing convincing data.

GENERAL GUIDELINES:

Text limits: Character count for a Tools paper is < 40,000, not including spaces. Count includes title page, abstract, introduction, results, discussion, and acknowledgments. Count does not include materials and methods, figure legends, references, tables, or supplemental legends.

Figures: Tools may have up to 10 main text figures. Figures must be prepared according to the policies outlined in our Instructions to Authors, under Data Presentation, <https://jcb.rupress.org/site/misc/ifora.xhtml>. All figures in accepted manuscripts will be screened prior to publication.

*****IMPORTANT:** It is JCB policy that if requested, original data images must be made available. Failure to provide original images upon request will result in unavoidable delays in publication. Please ensure that you have access to all original microscopy and blot data images before submitting your revision. ***

Supplemental information: There are strict limits on the allowable amount of supplemental data. Tools may have up to 5 supplemental figures. Up to 10 supplemental videos or flash animations are allowed. A summary of all supplemental material should appear at the end of the Materials and methods section.

Please note that JCB now requires authors to submit Source Data used to generate figures containing gels and Western blots with all revised manuscripts. This Source Data consists of fully uncropped and unprocessed images for each gel/blot displayed in the main and supplemental figures. Since your paper includes cropped gel and/or blot images, please be sure to provide one Source Data file for each figure that contains gels and/or blots along with your revised manuscript files. File names for Source Data figures should be alphanumeric without any spaces or special characters (i.e., SourceDataF#, where F# refers to the associated main figure number or SourceDataFS# for those associated with Supplementary figures). The lanes of the gels/blots should be labeled as they are in the associated figure, the place where cropping was applied should be marked (with a box), and molecular weight/size standards should be labeled wherever possible. Source Data files will be made available to reviewers during evaluation of revised manuscripts and, if your paper is eventually published in JCB, the files will be directly linked to specific figures in the published article.

The typical timeframe for revisions is three to four months. While most universities and institutes have reopened labs and

allowed researchers to begin working at nearly pre-pandemic levels, we at JCB realize that the lingering effects of the COVID-19 pandemic may still be impacting some aspects of your work, including the acquisition of equipment and reagents. Therefore, if you anticipate any difficulties in meeting this aforementioned revision time limit, please contact us and we can work with you to find an appropriate time frame for resubmission. Please note that papers are generally considered through only one revision cycle, so any revised manuscript will likely be either accepted or rejected.

Thank you for this interesting contribution to Journal of Cell Biology. You can contact us at the journal office with any questions at cellbio@rockefeller.edu.

Sincerely,

Kenneth Yamada, MD, PhD
Editor
Journal of Cell Biology

Dan Simon, PhD
Scientific Editor
Journal of Cell Biology

Full Revision

Manuscript number: JCB manuscript #202404003T (RC-2023--02239R)

Corresponding author(s): Hironobu, Fujiwara

[Please use this template only if the submitted manuscript should be considered by the affiliate journal as a full revision in response to the points raised by the reviewers.]

*If you wish to submit a preliminary revision with a revision plan, please use our "Revision Plan" template. **It is important to use the appropriate template to clearly inform the editors of your intentions.**]*

1. General Statements [optional]

This section is optional. Insert here any general statements you wish to make about the goal of the study or about the reviews.

The goals of our study are to (1) report the successful development of two fluorescently tagged *Col4a2* mouse models that fully support embryonic development and physiological functions, even when homozygous; (2) report real-time visualisation methods of COL4A2 and basement membrane structures in living developing organs; and (3) demonstrate the spatially different molecular and tissue dynamics of the basement membrane and its involvement in mammalian organogenesis. I appreciate that the goal of this study was acknowledged by the reviewers.

We would like to express our appreciation for the high-quality feedback received from our reviewers. We found the reviewers' comments to be both relevant and constructive, offering clear paths for improvement that are feasible within a reasonable timeframe and cost constraints, with two exceptions explained in the following sections. We have addressed these comments thoroughly in our revised manuscript.

Considering recent publications, we wish to acknowledge the contributions of a similar study establishing a mouse model, *mTurq2-Col4a1* (Jones et al., *Journal of Cell Biology* 2024, PMID: 37808687, doi: <https://doi.org/10.1101/2023.09.27.559396>). The work by Jones et al. offers valuable insights into the dynamics of mammalian basement membranes and represents a significant step forwards in the field. However, homozygous viability is compromised, as evidenced by the absence of homozygous pups from heterozygous crosses, suggesting that the fusion COL4A1 protein cannot substitute for endogenous COL4A1 functions. In contrast, our study introduced genes encoding a fluorescent protein in the *Col4a2* gene (and the *Col4a1* gene, although not discussed in our current study) in different regions, employed a different strategy, and demonstrated normal growth in homozygous mice. This distinction not only demonstrates the functional viability of the mouse model but also greatly broadens the scope for future research into embryonic development, physiological functions, and *in vitro* models such as organoids. Furthermore, our findings on dynamic basement membrane remodelling and its

Full Revision

involvement in hair follicle organogenesis, which were not addressed in the Jones et al. study, add new dimensions to the understanding of this process. In our revised manuscript, we referenced Jones et al., aiming to contextualise our findings within the broader landscape of basement membrane research (Page 18, line 21– Page 19, line 9).

Additionally, a new paragraph was added to the introduction to describe the role of the basement membrane in hair follicle development and the remaining questions regarding basement membrane dynamics (Page 6, lines 3–13). To meet the journal's word limit, we revised the title and abstract. We also refined the text through further English editing. However, except for the points indicated in the response below, these revisions do not impact the overall argument of the paper.

We believe these distinctions in mouse models and the scope of functional investigations offer substantial contributions to the field. We hope this clarification highlights the unique value and potential impact of our study.

This section is mandatory. Please insert a point-by-point reply describing the revisions that were already carried out and included in the transferred manuscript.

Reviewers' comments:

Reviewer #1

Summary:

Wuergzhen et al. generated two fluorescently tagged Col4a2 mouse models, including EGFP-Col4a2 and the photoconvertible mKikGR-Col4a2. They showed that tagged collagen IV get incorporated in the basement membranes of various tissues in the developing and adult mice, and fully support embryonic development, fertility, and physiological functions. The authors followed up investigating Col4a2 dynamics during embryonic development of the hair follicle. Using FRAP, they showed that the basement membrane expands with a faster rate near the tip region. This coincided with faster turnover rates of Col4a2 at the tip relative to the lower and upper stalk. In addition, the authors demonstrated that Col4a2 turnover depends on MMPs.

Response:

The reviewer referred to our mouse models as “EGFP-Col4a2” and “mKikGR-Col4a2”, thereby placing the names of the fluorescent proteins before Col4a2, suggesting that these designations follow the appropriate naming conventions. Thus, we consulted the RIKEN Bioresource Research Center about the proper naming of our mouse lines. In accordance with their guidance, we named them “eGFP-Col4a2” and “mKikGR-Col4a2”, respectively. We appreciate this reviewer's attention to this matter.

Full Revision

Major comments:

- Are the key conclusions convincing?

- The major conclusions of the manuscript are convincing. These include that tagging collagen IV does not compromise its function, differential expansion of the basement membrane, differential turnover of the Col4a2, the MMP dependence for normal basement membrane expansion and turnover. However, some claims detailed below need to be clarified or addressed to improve the manuscript.

- Should the authors qualify some of their claims as preliminary or speculative, or remove them altogether?

- In Figure 2B-C, the authors conclude that basement membrane expands at different rates, depending on the region with higher expansion rates near the hair follicle tip. From their methods, they conducted repetitive photobleaching cycles every 2-3 hours to maintain the bleached status of ROIs and tracked changes in basement membrane length. However, it sounds challenging to repetitively photobleach the same ROI in a dynamically expanding tissue, which may compromise the accuracy of length measurements. It would be helpful if the authors could provide movies corresponding to these experiments for clarity. In addition, this repetitive bleaching should be highlighted in the figure and figure legend, as readers could get confused about why there is no recovery here when comparing to the FRAP experiment.

Response:

Thank you for pointing this out. To show the process of repetitive photobleaching, we included supplementary figures (Fig. S2) and a detailed explanation in the Methods (Page 43, line 3–Page 44, line 6) and Figure Legends (Page 32, line 1–Page 33, line 4). We included snapshot images from the time lapse, as they provide a clearer explanation of the re-photobleaching process than the video does. Briefly, we re-photobleached at a timepoint (every 2–3 hours) in which the border of photobleached and nonphotobleached basement membrane regions remained distinguishable from the difference in their fluorescence intensities. We adjusted the ROI size and position for each photobleaching session according to the position of the border of photobleached and nonphotobleached basement membrane regions. Additionally, we included a note in the figure legend to indicate that re-photobleaching was performed. In the expanding basement membrane, it is challenging to label specific positions permanently because of the active turnover of its constituent proteins. Therefore, we believe that this method is currently considered a reasonable and reliable approach for observing and measuring local expansion of the basement membrane. We hope that new techniques that allow for the stable embedding of permanent markers in specific regions of the actively turning over and expanding basement membrane will be developed in the future.

- In Lines 180-198 and Figure 2F-H, the authors interpreted cell movement as contributed by cell-autonomous vs basement membrane expansion, which is speculative. Another possibility is that cell movement was all autonomous, while basement membrane expansion was caused by

Full Revision

mechanical stretching that was in turn caused by cell proliferation. This conclusion should be rephrased.

Response:

We appreciate the reviewer's insightful suggestion that cell movement might be entirely autonomous, with expansion of the basement membrane resulting from mechanical stretching due to cell proliferation. This perspective is discussed in the Discussion section of our revised manuscript (Pages 20, line 17–Page 22, line 3). The relevant sections were also revised by considering the overall tone and length of the Discussion. Moreover, we recognised the difficulty in defining the range of “cell autonomous” within the context of our research. The term “cell autonomous” generally refers to effects or behaviours that originate within a cell, independent of external influences or interactions with other cells. However, the complex interplay between cells and their microenvironment, particularly in a process such as basement membrane expansion, challenges the simplicity of this definition. Thus, we decided to refrain from using the term “cell autonomous” and rephrased our observations in the Results section (Page 10, lines 17–20).

- Following MMPi treatment, the authors found that basement membrane expansion was halted (Figure 4C-D). Yet, they later showed that hair follicles widened under MMPi treatment (Figure 4E-J). Under these conditions, I would have expected such widening to be accompanied by basement membrane expansion at the lower or upper stalk, which is not the case, at least for the first 7 hours (Figure 4C-D). How do the authors interpret this?

Response:

We appreciate the reviewer's valuable comment. As you noted, we observed that hair follicles widened under MMPi treatment, despite BM expansion being halted in the lower and upper stalk regions. Our new additional results suggest that this apparent discrepancy can be explained by the differential impact of MMPis on BM dynamics over time.

First, we acknowledge that our explanation and presentation of the MMPi experimental procedure and results were insufficient, making the findings difficult to interpret. In response, we have revised the manuscript to provide a clearer and more detailed description of the experimental procedure (Methods, Results, and Figure 4A) and have conducted additional live imaging experiments immediately after the addition of MMPis (Figs. 4E and Fig. 4G-O).

Here, we briefly explain the MMPi experimental procedure. In our original manuscript, skin explants were cultured for 16 hours after the addition of an MMPi, followed by a 20-hour live imaging session to ensure the uniform distribution of the effects of the MMPi throughout the tissue. However, upon evaluating our results in light of your comment, we found that hair follicle width had already increased at the start of this imaging period (Figs. 4P, R). Therefore, we analysed BM expansion during two distinct time periods: the first 7 hours immediately after MMPi addition and another 7 hours starting at 16 hours post-MMPi treatment.

In the early stages following MMPi treatment (0–7 h), we observed that BM expansion in the tip region was slightly reduced, whereas the BMs in the lower and upper stalk regions

Full Revision

tended to shrink further (Fig. 4E). This finding indicates that BM expansion can occur in some regions, whereas the spatial pattern of BM remodelling is altered. This altered pattern likely contributes to the widening of the hair follicle observed during this period. At the end of this first imaging period, we observed wider and shorter hair follicle structures (Figs. 4G–O).

As MMP inhibition progressed into the later stages (16–23 h posttreatment), both COL4A2 turnover and BM expansion were almost completely inhibited, and hair follicle shape changes were no longer observed (Figs. 4B–D, F, P–U). These findings suggest that initial, partial BM expansion with altered spatial patterns could cause hair follicle widening. However, once BM expansion is fully inhibited, hair follicle shape changes are also arrested.

In summary, we interpret that the widening of the hair follicle during MMP inhibition could be due to the early stage of partial BM expansion with an altered spatial pattern. As MMP inhibition progresses and BM expansion is almost entirely suppressed, further changes in hair follicle shape are inhibited. We believe that this explanation addresses the observed phenomenon and provides additional insights into the role of MMPs in regulating BM expansion and hair follicle morphogenesis (Page 13, line 7–Page 15, line 7).

- Would additional experiments be essential to support the claims of the paper? Request additional experiments only where necessary for the paper as it is, and do not ask authors to open new lines of experimentation.

- Considering the rapid expansion of BM near the tips, I wonder whether the authors have explored possible structural differences of the basement membrane in different regions? Is the basement membrane near the tips thinner and/or does it have microperforations, as reported in other systems? Looking into this may further support their observations, not only in control conditions but also in MMPi treated follicles.

Response:

To investigate potential differences in the thickness of the basement membrane in different regions, we used the eGFP-COL4A2 signal as a proxy for the basement membrane structure and measured its thickness in different tissue regions (Figs. 2D, E, Page 9, lines 11–15). We found that the eGFP-COL4A2 signal was significantly greater in the junctional/upper stalk regions than in the lower stalk and tip regions. There was no significant difference between the lower stalk and the tip. We also investigated the presence of micro-perforations via whole-mount 3D imaging of eGFP-COL4A2 (Fig. S3, Page 9, lines 15–24). We did not observe any clear mesh-like micro-perforations in any regions of the basement membrane but found pores in the basement membrane of the lower part of some hair follicles. Although we do not understand the functionality of these pores, we assume that these pores might be the temporal pores for the transmigration of cells such as melanocytes and immune cells through the BM. These pores were not observed after MMPi treatment, suggesting that MMP activity plays a role in their formation. These observations suggest organ-specific variations in basement membrane remodelling. We explain these new data in our revised manuscript as explained above.

Full Revision

- Are the suggested experiments realistic in terms of time and resources? It would help if you could add an estimated cost and time investment for substantial experiments.

- Yes, for possible structural changes, 3D rendering of fluorescence imaging as shown in Figure 1J may be sufficient.

- Increasing sample number using existing mouse strains should also be feasible. It would take 2-3 weeks from setting up timed pregnant mice to imaging the embryonic skin explants. Adding image analysis and figure preparation, it should be doable within 1 month.

Response:

We agree that increasing the sample size is important for enhancing the reliability of the results. However, a significant amount of cost and time investment are needed for the suggested replicate numbers (~10 total hair follicles) in our experimental system (long-term live imaging with mouse embryo tissues). Typically, high time- and spatial-resolution continuous 4D live imaging yields data from only one hair follicle per culture of embryonic skin tissue. This process requires the use of a two-photon microscope for ~24 hours. Thus, to obtain images of five hair follicles (yielding five biological replicates), for example, at least five timed mating events on five different days are needed, in alignment with the availability of the two-photon microscope. Occasionally, live imaging sessions must be terminated prematurely if the developing hair follicles move out of the imaging fields, particularly along the z-axis. Consequently, increasing the number of replicates for live imaging data demands a significant number of mouse mating and imaging sessions. Considering the cost and time investment for live imaging, we decided to end the acquisition of replicate data once we obtained high-quality data with low variability, even if this occurred before the number of replicates recommended by the reviewer was reached. In addition, some of the data with a high degree of variation resulted from a slight difference in the developmental stages of the hair follicles used rather than a variation in the trend of differences in the measurement results between hair follicle regions. Therefore, we mentioned this potential cause of variation in the revised manuscript and normalized some of the data (e.g., Figs. 4J, L).

- Are the data and the methods presented in such a way that they can be reproduced?

- In Figure 2D/ Figure 5C, it is unclear how ROIs corresponding to tip/lower/upper stalk are being drawn for quantification of Ki67+ cells.

Response:

For quantification of Ki67+ cells, the ROIs were defined based on the hair follicle's tissue architecture. The tip region was identified as the area between the bent junction connecting the tip and stalk, whereas the stalk region was defined as the area between the bent junction at the tip-stalk interface and the bent junction between the stalk and the interfollicular epidermis. The stalk region was further divided into upper and lower sections at the midpoint. This ROI

Full Revision

definition has now been explained in the Methods section (Page 41, line 24–Page 42, line 6) and is illustrated in Figure 2F. In response to Reviewer #2's comment, the data for quantifying proliferating cells (Ki67+ cell counting) were replaced with EdU+ cell counts (Page 10, lines 4–7).

- *The images in 2B and 4C, 3A and 4E, are identical. Images should not be re-used in figures. This raised the concern that probably not sufficient samples were imaged to have different representative images. It should also be clarified where the data was re-used, for example, the control data in 2C and 4D, 3B and 4B.*

Response:

We appreciate the reviewer's suggestion and have replaced all the reused images with different images (Fig. 2B and 4P). Notably, the original manuscript acknowledged the reuse of certain data. This approach was taken to maintain consistency across related experiments, although we understand and agree with the reviewer's concern.

- Are the experiments adequately replicated and statistical analysis adequate?

- *A few experiments have low sample numbers while the data was quite variable. These include Figure 2G-H (n = 3 cells), Figure 3C (unknown sample number), Figure 4B (n = 3 hair follicles), Figure 5B (n = 3 hair follicles). More sample numbers (~10 total) should be included to solidify the findings.*

Response:

We have conducted additional experiments to increase the number of replicates. These include Fig. 2I–J (n = 7 hair follicles), Fig. 3D (n = 3 hair follicles), Fig. 4C, D (n = 6 hair follicles), Fig. 5B (n = 6 hair follicles), and Fig. 5F, G (n = 8–11 hair follicles). We also ensured that an adequate sample size was obtained for the newly added experiments. On the basis of the reasons described above, we performed data acquisition once we obtained high-quality data with low variability, even if this occurred before the number of replicates recommended by the reviewer was reached. We also added a new Fig. 5G to effectively present the data with increasing sample numbers.

- *For the mKikGR experiment in Figure 3C, quantification should be included. A ratiometric measurement of green/red fluorescence over time should be a good complementary way of demonstrating region-specific recovery.*

Response:

We attempted ratiometric measurement of green/red fluorescence for the experiment shown in Figure 3C via single z-plane images. However, we encountered significant variability due to the

Full Revision

low fluorescence intensity, low signal-to-noise ratio, and high susceptibility of mKikGR to photobleaching. As a result, instead of presenting quantification data obtained from these conditions, we opted to include replicate maximum projection images in Figure S5 to better illustrate region-specific recovery. In our experimental system, long-term measurement of the turnover of basement membrane proteins with photoconvertible proteins remains a significant challenge and requires improved imaging tools and techniques.

- In Figure 5B, the authors claim that the percentage of dividing cells in control follicles being different from that of MMPi-treated follicles. How are they extracting these percentages? From the plots, control and MMPi-treated columns do not appear to be normalized as 100% to make such comparisons. Moreover, having two mean \pm SD in each column makes these data confusing to interpret. The authors should consider replotting their data either by combining their data into a unique population per conditions and reporting the percentages, or alternatively, they may consider splitting each of these columns into two (i.e., dividing vs. non-dividing cells), and comparing both conditions as ratios of dividing versus/non-dividing cells.

Response:

We increased the sample number and replotted the data to show the frequency of cell division per hour under each condition (Fig. 5B, Page 15, lines 15–16).

Minor comments:

- Specific experimental issues that are easily addressable.
- *Already addressed above.*

- Are prior studies referenced appropriately?
- *I believe so.*

- Are the text and figures clear and accurate?
- *All plots measuring basement membrane length are labeled as 'increase in BM length', even when in some cases the BM length is reduced. The authors should consider relabeling these as 'BM length change' or something similar.*

Response:

We relabelled them as 'BM length change'.

- *The second and third paragraphs of the discussion are too long and should be condensed into a single paragraph.*

Response:

We condensed these two paragraphs into a single paragraph.

Full Revision

- Do you have suggestions that would help the authors improve the presentation of their data and conclusions?
- *Use better color combination for visualization of multi-channel images. For instance, magenta/green is a better combination than red/green for the color blind. Color combinations are not consistent across figures in the manuscript.*

Response:

We changed the colour combinations to magenta/green for all the figures.

- *Include movies for all data derived from live imaging.*

Response:

We have submitted all the data derived from live imaging to the SSBD public bioimage database (<https://ssbd.riken.jp/database/>).

- *Include statistical tests used for all plots, some are missing.*

Response:

We included the statistical tests employed for all plots.

- *The authors should consider fitting their FRAP data in each condition and report percentages corresponding to the mobile and immobile fractions.*

Response:

Thank you for your advice. After careful consideration, we concluded that calculating the mobile and immobile fractions in our FRAP analysis was not feasible for our experimental system. Our FRAP observations indicated very slow signal recovery and the lack of a clear plateau within the experimental timeframe (Figure 3B). This slow recovery, coupled with dynamic changes in the tissue structure and potential increases and decreases in the total amount of eGFP-COL4A2 in the basement membrane, underscores the nonstatic nature of our experimental system. These factors introduce significant variability in molecular dynamics as a whole system, thereby making the application of standard fitting models, which presuppose stable conditions, inappropriate for our data.

Moreover, the unique protein dynamics of COL4A2, including its limited diffusibility and slow turnover rates within the basement membrane, necessitate prolonged live imaging sessions, potentially ranging from hours to days, to capture the full recovery process. However, maintaining constant tracking of the photobleached basement membrane zone throughout this period becomes increasingly challenging owing to morphological changes in embryonic skin

Full Revision

tissues, including those of the basement membrane. Furthermore, as the fluorescence recovery rate exceeds approximately 50%, distinguishing the boundary between the bleached and nonbleached areas becomes difficult, complicating accurate measurements.

Given these challenges, particularly the difficulty in maintaining continuous measurements to achieve a recovery plateau, we opted not to calculate the mobile and immobile fractions. To ensure consistency of photobleaching across target ROIs, we applied the same laser irradiation programme for photobleaching within each ROI from different regions (Page 42, line 23–Page 43, line 19). Furthermore, to delineate the significant difference in signal intensity recovery between regions, we extracted data from specific timepoints and presented this subset separately to highlight statistically significant differences (Figure 3C).

- *Examples of horizontal vs perpendicular cell division appear to be mislabeled in Video5.*

Response:

We corrected this error accordingly.

Reviewer #1 (Significance (Required)):

- Describe the nature and significance of the advance (e.g. conceptual, technical, clinical) for the field.

- *The large molecular weight of extracellular proteins makes it challenging to generate genetically engineered versions of such proteins to investigate their function and dynamics in vivo. The present study has addressed such issues for Col4a2, a major component of the basement membrane. This study further provides insights towards the understanding of BM dynamics during embryonic organ development.*

- Place the work in the context of the existing literature (provide references, where appropriate).

- *Addressed above*

- State what audience might be interested in and influenced by the reported findings.

- *Cell and developmental biology, extracellular matrix biology, organ development and regeneration.*

- Define your field of expertise with a few keywords to help the authors contextualize your point of view. Indicate if there are any parts of the paper that you do not have sufficient expertise to evaluate.

- *Tissue morphogenesis, extracellular matrix*

Reviewer #2

Full Revision

The authors successfully generated a knock-in mouse line expressing fluorescence-tagged endogenous collagen IV. Homozygous mice exhibited normal survival with no apparent developmental abnormalities. Through imaging of the back skin of COL4A2-eGFP mice in ex vivo culture, the authors observed a faster expansion rate of the basement membrane (BM) at the growing tips of developing hair follicles, while the upper stalk region showed a slower BM expansion rate. Real-time imaging also revealed that the BM and basal epithelial cells moved in the same direction. Fluorescence recovery after photobleaching (FRAP) and the photoconvertible protein mKikGR experiments demonstrated a faster turnover rate of COL4A2 at the tips of the hair follicles. The authors used MMP inhibitors to suppress the recovery of COL4A2-eGFP, BM expansion, and observed abnormalities in the developmental process of hair follicle morphology. The findings are novel and interesting, some aspects of how the experiments and quantifications were conducted should be clarified to allow the readers to fully appreciate the results.

Major Comments:

1. BM Expansion Statistics: Regarding the statistical analysis of BM expansion, is the length of the BM affected by the z-axis plane at the time of imaging? How is it ensured that the BM length measured at different time points corresponds to the same z-axis?

Response:

Thank you for pointing this out. To ensure accurate and consistent measurement of the basement membrane length over time, we carefully selected and monitored the appropriate z-plane at each time point. We have provided detailed visual explanations in our revised manuscript (Fig. S2).

To maintain consistency, we selected a z-plane characterized by clear and sharp basement membrane signals, indicating that the basement membrane was oriented vertically in the targeted area, which was near the midline of the hair follicle. On the other hand, tilted basement membranes off-midline can be identified as unclear and blurred basement membrane signals. This selection ensured that we measured the same tissue region over time.

For photobleaching, we targeted 10 ± 3 continuous z-planes (at $1 \mu\text{m}$ intervals) to capture the basement membrane as clear line-like structures on different z-planes, creating a 3D cuboidal bleached area. The same photobleaching process was applied 2–3 times. After 7 h, the z-planes where the clear basement membrane of each compartment could be observed were selected for BM length measurement, as shown in Figure S2B. As no significant deformations or morphological changes were observed at the boundaries of the 3D cuboidal bleached area during the imaging period, as shown in Figure S2C, we conclude that the basement membrane maintains a consistent shape and expansion rate, even with shifts in the circumferential location of the hair follicle. This suggests that any potential displacement around the circumference does not affect the accuracy of the measurements.

Therefore, our approach allowed us to track and measure the basement membrane length changes accurately over time, ensuring that our data reflect the true expansion dynamics

Full Revision

of the basement membrane without interference from changes in the circumferential location or z-plane alignment.

2. *Cell Movement Statistics (Figure 2): In the statistical analysis of cell movement in Figure 2, the lack of other references raises questions about ensuring that the cells measured at different time points are the same cells.*

Response:

To demonstrate that the same cells can be accurately tracked over time in our movie, we included corresponding time-lapse images in Figure S4.

3. *The authors concluded that the inhibition of MMP delayed the recovery of COL4A2-eGFP after photobleaching, indicating the crucial role of MMP activity in the incorporation of COL4A2 into the BM. Does MMP inhibition lead to a reduction in the synthesis of the COL4A2 protein, thereby delaying the recovery of COL4A2 in the BM?*

Response:

As it is difficult to directly measure the amount of cellular synthesis of the COL4A2 protein *in vivo*, we used RNAscope to examine changes in *Col4a2* mRNA expression levels under MMPi conditions as a proxy for COL4A2 protein synthesis (Figure S6, Page 13, lines 13–14). Our analysis did not reveal any significant changes in *Col4a2* mRNA expression levels under MMPi conditions. Therefore, we conclude that the delayed recovery of eGFP-COL4A2 observed after photobleaching may not be due to reduced synthesis of the COL4A2 protein. Instead, these findings suggest the essential role of MMP activity in incorporating COL4A2 into the basement membrane.

Minor Comments:

In Figure 1D, the representative image and the statistical graph are inconsistent. For example, in the upper stalk region, all basal epithelial cells have Ki67 signals according to the representative image.

Response:

Thank you for bringing this to our attention. We acknowledge that the percentage of Ki67-positive cells depicted in the statistical graph of the original Figure 2E shows significant variability, and there appears to be a lack of consistency between the representative image in the original Figure 2D and the statistical graph. This inconsistency may be attributed to the variable signal intensities observed in our Ki67 staining, including instances of very faint staining, which could lead to variability in the detection of Ki67-positive cells across samples. Furthermore, since Ki67 is expressed throughout nearly the entire mitotic process, it does not

Full Revision

precisely indicate proliferating cells within a specific time frame. To compare regions with high mitotic activity more accurately and measure cell division rates, we conducted EdU incorporation assays and increased the number of replicates. We found that although EdU+ cells were present throughout the hair follicles, the basal epithelial cells in the tip and lower stalk regions contained approximately twice as many EdU+ cells as those in the upper stalk region (Figure 2F, G, Page 10, lines 4–7).

Reviewer #2 (Significance (Required)):

The authors successfully generated knock-in mice expressing fluorescence-tagged endogenous collagen IV. With this novel mouse model, researchers can directly observe the dynamic changes in the extracellular matrix of different organs in mammals during development and disease, revealing the potential roles of the extracellular matrix.

Reviewer #3

Reviewer #3 (Evidence, reproducibility and clarity (Required)):

Abe et al. generated multiple novel mouse model that follows and labels Col4a2 by fusing it with GFP. The creation of this mouse model allowed for the analysis of the basement membrane in real-time using live imaging of embryonic explant cultures. The combination of the mouse model with live imaging revealed new insights into how the basement membrane turnover occurs during early hair follicle development. After setting the stage the authors establish its utility with embryonic tissue, which revealed membrane buds that could be visualized with eGFP at different time intervals for 19h. The growth of the hair follicle bud during that time could be divided three distinct areas, the upper, lower stalks, and the tip. Where the tip and lower stalk increased in BM length, while the upper stalk decreased. BM length correlated with increased cellular replication.

Using a double transgenic model system, they evaluated the differences between 'cell displacement', 'BM expansion', and 'cell autonomous movement'. Diving deeper with the Col4a2-mKiGR mouse model investigated Col4a2 turnover to reveal similar results mentioned above. Lastly MMP inhibitors were used to investigate how hair follicle growth is affected, which revealed inhibition of cellular replication and inhibition of Col4a2 turnover. These results inhibited hair follicle elongation which made the forming hair follicles thicker.

Overall, this is an interesting paper for hair follicle biologists since it investigates the early moments of the budding follicle in real time using a ex vivo culture model. In addition, there is additional appeal because the authors use live imaging the hair follicle neogenesis as a basement membrane model, which is interesting and novel. The manuscript is clearly written, and the data support the overall conclusions. My comments below are to help in clarity which may be used to develop clearer figures and additional text.

Full Revision

Major:

In Figure 2 for the cell displacement and expansion part. I found the figure and text confusing, particularly in regard to how the data shows the bleached edge of the BM moving alongside the cells. Could the authors make this clearer in the figure and in the writing of the text?

Response:

To illustrate how the bleached edge of the basement membrane moves alongside the cells, we have included corresponding time-lapse images in Figure S4. These images demonstrate that both the same cells (marked with yellow dots) and the tip of the photobleached edge (indicated by white arrowheads) can be accurately tracked over time. We performed this tracking using 3D live imaging data captured at approximately 30-min intervals. This high spatial and temporal resolution enables accurate tracking of cell movements and photobleached edges over time.

The authors created a Col4a2-mKikGR mouse line that is supposed to follow Col4a2 turnover. It is difficult to understand how the authors can claim that it was turnover with the explanation in the text of how the mouse model works. Could the authors write a better description of the mouse model in the text?

Response:

We provided an explanation of the mKikGR mouse model and the experimental procedures that enabled us to assess the turnover of COL4A2 (Page 12, lines 2–4). Briefly, this mouse model allows us to measure the dynamics of both pre-existing mKikGR-COL4A2 proteins (photoconverted to red (magenta)) and newly incorporated mKikGR-COL4A2 proteins (green) in a specific region over time. For example, in the lower stalk region of the BM, the magenta signal gradually decreased and almost disappeared during 8–10 hours of culture, whereas the green fluorescence gradually increased and replaced the red (magenta) signal (Fig. 3D, Supplementary Video 3 and Fig. S5). This red (magenta)-to-green colour change indicated turnover of the COL4A2 protein, where pre-existing mKikGR-COL4A2 proteins were replaced by newly synthesized or recruited proteins.

Reviewer #3 (Significance (Required)):

The strengths of the paper are the use of a novel mouse model that can track collagen (Col4a2) with a tagged GFP and the live imaging model system. This has led to a novel and important knowledge on the interplay between the BM and cells. The limitation of the manuscript is that the entire manuscript utilizes a single protocol (live imaging) to investigate BM dynamics. This might be due to the sophisticated nature of the model systems, but it is a limitation.

To my knowledge the study is novel because of the mouse model to track Col4a2 dynamics with GFP. This model led to some interesting findings about hair follicle development in that different regions expand differently with the BM. This is good foundational knowledge using the latest

Full Revision

state of the art techniques in live imaging of embryonic tissue culture systems.

The audience that would be interested in this manuscript is broad, which would span cell and developmental biology but also specialists in membrane and protein biology that can finally see in real time the dynamics of Col4a2 building skin and hair.

My expertise is in hair follicle development and repair and stem cell biology.

October 30, 2024

RE: JCB Manuscript #202404003R

Dr. Hironobu Fujiwara
RIKEN Center for Biosystems Dynamics Research
2-2-3 Minatojima Minamimachi
Chuo-ku
Kobe 6500047
Japan

Dear Dr. Fujiwara,

Thank you for resubmitting your manuscript entitled "eGFP-Col4a2 mouse model reveals basement membrane dynamics underlying hair follicle morphogenesis." We find that your extensive and conscientious revisions have fully resolved, within reasonable limits of research practicality, the concerns of the expert reviewers.

We did, however, identify one minor but potentially significant concern in our accelerated re-reviewing process: in your figures with bar graphs comparing more than a pair of conditions, specifically in Figures 2-5 and S6, you applied two-tailed unpaired t-tests to compare each of two different conditions amongst groups of three or more conditions. Please clarify whether in each case you applied the Bonferroni correction to standard t-tests to correct for the fact that your graphs are not comparing a simple pair of conditions. If not, please do so, or perhaps ideally apply ANOVA with an appropriate post-hoc test, to ensure that the statistical significance values listed in the figures are correct. If the reason for this concern is not clear, please consult statistics textbooks, Wikipedia, or professional statisticians.

We hope that you can resolve this remaining concern about this substantially strengthened, now-excellent and informative manuscript promptly so that we can proceed to final acceptance. We would be happy to publish your paper in JCB pending final revisions necessary to address the above mentioned concern and to meet our formatting guidelines (see details below).

A. MANUSCRIPT ORGANIZATION AND FORMATTING:

1) Text limits: Character count for Tools is < 40,000, not including spaces. Count includes title page, abstract, introduction, results, discussion, and acknowledgments. Count does not include materials and methods, figure legends, references, tables, or supplemental legends.

2) Figure formatting: Tools may have up to 10 main text figures. Scale bars must be present on all microscopy images, including inset magnifications. Molecular weight or nucleic acid size markers must be included on all gel electrophoresis. Please avoid pairing red and green for images and graphs to ensure legibility for color-blind readers. If red and green are paired for images, please ensure that the particular red and green hues used in micrographs are distinctive with any of the colorblind types. If not, please modify colors accordingly or provide separate images of the individual channels.

3) Statistical analysis: Error bars on graphic representations of numerical data must be clearly described in the figure legend. The number of independent data points (n) represented in a graph must be indicated in the legend. Please, indicate whether 'n' refers to technical or biological replicates (i.e. number of analyzed cells, samples or animals, number of independent experiments). If independent experiments with multiple biological replicates have been performed, we recommend using distribution-reproducibility SuperPlots (please see Lord et al., JCB 2020) to better display the distribution of the entire dataset, and report statistics (such as means, error bars, and P values) that address the reproducibility of the findings.

Statistical methods should be explained in full in the materials and methods. For figures presenting pooled data the statistical measure should be defined in the figure legends. Please also be sure to indicate the statistical tests used in each of your experiments (both in the figure legend itself and in a separate methods section) as well as the parameters of the test (for example, if you ran a t-test, please indicate if it was one- or two-sided, etc.). Also, if you used parametric tests, please indicate if the data distribution was tested for normality (and if so, how). If not, you must state something to the effect that "Data distribution was assumed to be normal but this was not formally tested."

4) Title: Please add "An" at the beginning of the title.

- 5) Materials and methods: Should be comprehensive and not simply reference a previous publication for details on how an experiment was performed. Please provide full descriptions (at least in brief) in the text for readers who may not have access to referenced manuscripts. The text should not refer to methods "...as previously described."
- 6) For all cell lines, vectors, constructs/cDNAs, etc. - all genetic material: please include database / vendor ID (e.g., Addgene, ATCC, etc.) or if unavailable, please briefly describe their basic genetic features, even if described in other published work or gifted to you by other investigators (and provide references where appropriate). Please be sure to provide the sequences for all of your oligos: primers, si/shRNA, RNAi, gRNAs, etc. in the materials and methods. You must also indicate in the methods the source, species, and catalog numbers/vendor identifiers (where appropriate) for all of your antibodies, including secondary. If antibodies are not commercial, please add a reference citation if possible.
- 7) Microscope image acquisition: The following information must be provided about the acquisition and processing of images:
- Make and model of microscope
 - Type, magnification, and numerical aperture of the objective lenses
 - Temperature
 - Imaging medium
 - Fluorochromes
 - Camera make and model
 - Acquisition software
 - Any software used for image processing subsequent to data acquisition. Please include details and types of operations involved (e.g., type of deconvolution, 3D reconstitutions, surface or volume rendering, gamma adjustments, etc.).
- 8) References: There is no limit to the number of references cited in a manuscript. References should be cited parenthetically in the text by author and year of publication. Abbreviate the names of journals according to PubMed.
- 9) Supplemental materials: Tools generally may have up to 5 supplemental figures and 10 videos. You currently exceed this limit but, in this case, we will be able to give you the extra space. Please also note that tables, like figures, should be provided as individual, editable files. A summary of all supplemental material should appear at the end of the Materials and methods section. Please include one brief sentence per item.
- 10) Video legends: Should describe what is being shown, the cell type or tissue being viewed (including relevant cell treatments, concentration and duration, or transfection), the imaging method (e.g., time-lapse epifluorescence microscopy), what each color represents, how often frames were collected, the frames/second display rate, and the number of any figure that has related video stills or images.
- 11) eTOC summary: A ~40-50 word summary that describes the context and significance of the findings for a general readership should be included on the title page. The statement should be written in the present tense and refer to the work in the third person. It should begin with "First author name(s) et al..." to match our preferred style.
- 12) Conflict of interest statement: JCB requires inclusion of a statement in the acknowledgements regarding competing financial interests. If no competing financial interests exist, please include the following statement: "The authors declare no competing financial interests." If competing interests are declared, please follow your statement of these competing interests with the following statement: "The authors declare no further competing financial interests."
- 13) A separate author contribution section is required following the Acknowledgments in all research manuscripts. All authors should be mentioned and designated by their first and middle initials and full surnames. We encourage use of the CRediT nomenclature (<https://casrai.org/credit/>).
- 14) ORCID IDs: ORCID IDs are unique identifiers allowing researchers to create a record of their various scholarly contributions in a single place. Please note that ORCID IDs are required for all authors. At resubmission of your final files, please be sure to provide your ORCID ID and those of all co-authors.
- 15) JCB requires authors to submit Source Data used to generate figures containing gels and Western blots with all revised manuscripts. This Source Data consists of fully uncropped and unprocessed images for each gel/blot displayed in the main and supplemental figures. Since your paper includes cropped gel and/or blot images, please be sure to provide one Source Data file for each figure that contains gels and/or blots along with your revised manuscript files. File names for Source Data figures should be alphanumeric without any spaces or special characters (i.e., SourceDataF#, where F# refers to the associated main figure number or SourceDataFS# for those associated with Supplementary figures). The lanes of the gels/blots should be labeled as they are in the associated figure, the place where cropping was applied should be marked (with a box), and molecular weight/size standards should be labeled wherever possible. Source Data files will be directly linked to specific figures in the published article.

Source Data Figures should be provided as individual PDF files (one file per figure). Authors should endeavor to retain a

minimum resolution of 300 dpi or pixels per inch. Please review our instructions for export from Photoshop, Illustrator, and PowerPoint here: <https://rupress.org/jcb/pages/submission-guidelines#revised>

16) Journal of Cell Biology now requires a data availability statement for all research article submissions. These statements will be published in the article directly above the Acknowledgments. The statement should address all data underlying the research presented in the manuscript. Please visit the JCB instructions for authors for guidelines and examples of statements at (<https://rupress.org/jcb/pages/editorial-policies#data-availability-statement>).

B. FINAL FILES:

Thank you for your attention to these final processing requirements. Please revise and format the manuscript and upload materials within 7 days. If you need an extension for whatever reason, please let us know and we can work with you to determine a suitable revision period.

Thank you for this interesting contribution, we look forward to publishing your paper in Journal of Cell Biology.

With kind regards,

Kenneth M. Yamada, MD, PhD
Senior Editor
Journal of Cell Biology

Dan Simon, PhD
Scientific Editor
Journal of Cell Biology